# Immunoaffinity Extraction and Alternative Approaches for the Analysis of Toxins in Environmental, Food or Biological Matrices

**DOI:** 10.3390/toxins12120795

**Published:** 2020-12-13

**Authors:** Nathalie Delaunay, Audrey Combès, Valérie Pichon

**Affiliations:** 1Department of Analytical, Bioanalytical Sciences and Miniaturization (LSABM), CBI ESPCI Paris, PSL University, CNRS, 75005 Paris, France; nathalie.delaunay@espci.fr (N.D.); audrey.combes@espci.fr (A.C.); 2Department of Chemistry, Sorbonne University, 75005 Paris, France

**Keywords:** toxins, immunoaffinity, immunosorbent, molecularly imprinted polymers, aptamers, oligosorbents, trace analysis, complex samples, matrix effects

## Abstract

The evolution of instrumentation in terms of separation and detection allowed a real improvement of the sensitivity and analysis time. However, the analysis of ultra-traces of toxins in complex samples requires often a step of purification and even preconcentration before their chromatographic analysis. Therefore, immunoaffinity sorbents based on specific antibodies thus providing a molecular recognition mechanism appear as powerful tools for the selective extraction of a target molecule and its structural analogs to obtain more reliable and sensitive quantitative analysis in environmental, food or biological matrices. This review focuses on immunosorbents that have proven their efficiency in selectively extracting various types of toxins of various sizes (from small mycotoxins to large proteins) and physicochemical properties. Immunosorbents are now commercially available, and their use has been validated for numerous applications. The wide variety of samples to be analyzed, as well as extraction conditions and their impact on extraction yields, is discussed. In addition, their potential for purification and thus suppression of matrix effects, responsible for quantification problems especially in mass spectrometry, is presented. Due to their similar properties, molecularly imprinted polymers and aptamer-based sorbents that appear to be an interesting alternative to antibodies are also briefly addressed by comparing their potential with that of immunosorbents.

## 1. Introduction

Given their presence at the trace level in food, biological or environmental samples, the analysis of toxins requires very sensitive and specific tools for their monitoring in very complex samples. For many years, liquid chromatography (LC) coupled with fluorescence detection (Fluo) has been used to monitor toxins with or without derivation steps depending on the physico-chemical properties of the targeted molecules—these toxins being mainly mycotoxins monitored in food matrices. Then, LC coupled with mass spectrometry (MS) has gradually become the preferred method to confirm the presence of these mycotoxins but also of other toxins at ultra-trace level in complex extracts. It also became the method of choice to eliminate the derivation step and allow the simultaneous monitoring of toxins of different classes that may be present in the same sample. However, when applied to the analysis of very complex extracts, LC–MS suffers from matrix effects during the ionization step that can lead to erroneous quantification. Thus, although initially developed to improve the reliability of less specific analytical methods such as LC-Fluo by allowing selective cleaning of sample extracts, immunoaffinity sorbents are still developed and used in combination with LC/MS to reduce or even eliminate matrix effects.

These immunoaffinity supports, also named immunosorbents (ISs), are based on the use of antibodies specific to the molecule(s) of interest. The high specificity and affinity of the antigen-antibody interactions allow the selective and efficient extraction of the target analyte(s) from complex samples, thus facilitating its final identification and quantification [1,2,3,4,5]. As a result, ISs are marketed as single-use cartridges and widely used for the monitoring of mycotoxins in foodstuffs. ISs are also still under development to propose new extraction formats such as magnetic beads to perform selective solid phase extraction (SPE) in dispersive mode (dSPE). ISs are also still under development for other types of toxins such as marine or plant toxins that need to be detected at trace levels in environmental matrices but also in biological fluids.

A molecular recognition mechanism can also be implemented using molecularly imprinted polymers (MIPs), whereby synthesis leads to the formation of specific cavities mimicking the recognition site of antibodies [6,7]. Another selective support, called oligosorbent (OS), has been also recently developed using aptamers immobilized onto a solid support. Aptamers are oligonucleotides with a specific sequence able to bind a given molecule with the same affinity as antibodies. OSs were recently successfully applied to the selective extraction of different target analytes from biological fluids and food samples [2,8,9]. Once the sequence is available, the development of an OS is less expensive than for an IS. MIPs and OSs present the advantage to be synthesized in a few days. In return, their application to real samples necessitates a careful optimization of the extraction procedure to reach the expected affinity and selectivity, while this selective procedure is very easy to develop when using ISs.

This review mainly focuses on the potential of ISs that have proven their efficiency to selectively extract various types of toxins of various sizes (from small mycotoxins to large proteins) and physicochemical properties. A large number of studies dealing with the use of ISs for toxin analysis are listed in this review and illustrate the high potential of these sorbents to lead to reliable quantitative methods, particularly in LC–MS, without systematically requiring tedious and costly calibration approaches, such as matrix match calibration, as matrix effects can be strongly reduced. However, the constraints in terms of extraction conditions are also highlighted. In addition, through an almost exhaustive inventory of work related to MIPs, some of which being already marketed thus offering an alternative to ISs for certain molecules, and to OSs, this review also highlights the potential of these alternative approaches but also their limitations in terms of development and applications. 

## 2. Immunoaffinity Sorbents

### 2.1. Antibody Production and Development of Immunosorbents

ISs were first described in the biological field because of the availability of antibodies specific to large molecules such as proteins. Indeed, small molecules (<1000 Da) are unable to evoke an immune response and make the production of antibodies more difficult. They have to be bound to a larger carrier molecule, usually a protein, to immunize the animal. After a few weeks or months of immunization by this immunoconjugate, the serum is collected and the antibodies, i.e., the immunoglobulin G (IgG) fraction, are purified. This purification generally results in polyclonal antibodies (pAbs) made of a heterogeneous mixture of antibodies [10,11]. These pAbs can bind with the antigen with different affinities, because they are directed against various antigenic determinants (epitopes) on the antigen/immunoconjugate. Therefore, when a small molecule is targeted, it is commonly found in the literature that the mix of pAbs contains only about 15% active antibodies [10]. In return, techniques of hybridoma allow the production of only one type of IgG known as monoclonal antibodies (mAbs). Polyclonal antibodies are cheaper to obtain, but their production suffers from a lack of reproducibility in terms of time of response of an animal, of quantity and even of specificity and ethical issues. In contrast, the production of mAbs is costly but guarantees a long-term production of reproducible antibodies that do not require animals for further large-scale production. Once the antibodies are obtained, they are immobilized on a solid sorbent, called IS.

The first ISs that were developed for toxin analysis targeted mycotoxins, despite the difficulty to produce specific Abs for such small molecules. Indeed, their quantification at low concentration levels, mainly in foodstuff, represents an analytical challenge. Therefore, a considerable effort has led to the development of mycotoxin-specific ISs, which are currently marketed by many companies such as Vicam and R-Biopharm and to a lesser extent RomerLabs, Aokin, and Neogen, as shown in Table 1. They have been mostly developed for the four aflatoxins (AFs) (B_1_, B_2_, G_1_, and G_2_), for Ochratoxin A (OTA), for trichothecene toxins such as deoxynivalenol (DON), zearalenone (ZON), H-2 and HT-2 toxins, fuminosins (FUM B_1_, B_2,_ B_3_) and for sterigmatocystin (SMC). As illustrated in this table, while some of these ISs have been developed to trap a mycotoxin and possibly its structural analogs, some companies sell cartridges containing two or more antibodies to simultaneously trap multiple toxins and their structural analogs. As an example, the AlfaOchra Test cartridge allows the trapping of the four main aflatoxins and OTA simultaneously or the Myco6in1 of the four AFs, OTA and several trichothecenes, including their metabolites such as nivalenol (NIV) and acetyl-deoxynivalenol (ADON). 

In addition to Table 1, Table 2 is a fairly exhaustive list of the work related to the development of ISs in academic research laboratories for toxin analysis. One third of this table is devoted to the development of ISs for mycotoxins but with new targets such as α- and β-amanitins, zeranol (ZER) and sterigmatocystin (SMC). The rest is devoted to the development of ISs for cyanotoxins (i.e., Anatoxin-A and microcystins (MCs)), diarrheic shellfish poisoning (DSP) toxins (such as okadaic acid (OA) and some dinophysistoxins (DTXs)), paralytic shellfish poison (PSP) toxins (such as saxitoxin (STX)), and even insect, bacterial or plant toxins that have the particularity to be protein toxins.

While the nature of the solid phase used to immobilize antibodies is rarely described for commercially available ISs, the content of Table 2, dedicated to ISs developed in research laboratories, illustrates the wide variability in the nature of the possible sorbent and the final format of the immunoextraction device. Indeed, if agarose gel such as Sepharose is one of the most widely used sorbents for developing commercially available ISs and is still used to develop new ISs (as shown by 40% of the work reported in Table 2), other sorbents can be used, such as activated silica or polymers. These supports, available as beads, can be packed between two frits in a conventional SPE cartridge such as the commercially available ISs or can be directly introduced into the sample to perform the extraction in dispersive mode as discussed later. To immobilize Abs on a solid sorbent, the most common approach is with regard to their covalent bonding, which is often achieved by coupling an accessible amino group of the Abs with a support that contains reactive groups such as epoxy [93] or aldehyde [97] or groups that can be activated using glutardialdehyde [71,100,101], carbonyldiimidazole, cyanogen bromide (CNBr) or N-hydrosuccinimide (NHS). Some activated supports are commercially available such as NHS- [70] or CNBr- [69,76,84,85,86,87,88,91,94,95,96,103,112] activated Sepharose or glutardialdehyde activated silica [74,77,82]. Non-covalent binding can also be used to couple Abs to the sorbent. For this purpose, proteins A- [90] or G- [80,99,104,105,106,107,108] based sorbents or sorbents grafted with anti-IgG [92,98] can be used as these proteins bind a part of the constant region of Abs, allowing the orientation of the Abs with the antigen binding sites away from the surface and towards the solution. The same orientation effect can be obtained using streptavidin activated sorbent that can react with biotinylated Abs [72,79,109]. However, the resulting non-covalent binding is quite strong under physiological conditions but can be easily disrupted by decreasing the pH of the surrounding solution. The sol–gel method can also be used to entrap Abs [73,89]. In this case, Abs are then immobilized in the pores of a hydrophilic glass matrix that reduces the non-specific adsorption of apolar analytes. Moreover, narrow pores prevent the diffusion of large size molecules such as bacteria or proteolytic enzymes. Therefore, no bacteriostatic agent must be added in the phosphate buffer saline (PBS) solution for IS storage [89].

### 2.2. Immunoextraction Procedure on IS Cartridges

Numerous studies reported the use of commercially available ISs for toxin analysis in numerous samples, and in some cases more than 100 samples were analyzed [36,43,47,58,59,67]. In most of the cases, the immunoextraction procedure provided by the vendor was directly applied by the user in terms of washing and elution conditions. For laboratory-made ISs, such as those reported in Table 2, both steps must be optimized but are very similar to those applied to commercially available ISs, with a washing step using water or a buffer and elution mainly with methanol. In addition to aqueous conditions, a low amount of solvent [71,84,89,95] (or surfactant [102,108] for proteins) can be added in the washing solution to improve the selectivity by removing the interfering compounds retained by non-specific interactions mainly caused by the solid-phase selected for the grafting. To evaluate the contribution of non-specific interactions in the retention of the target toxins, we proposed to compare their retention on their IS with their retention on a sorbent bonded with non-specific antibodies [71,107] or on a non-bonded sorbent [99]. The study of the retention on the IS of compounds having a polarity similar to one of the target toxins, but which should not be retained as they are not recognized by antibodies, was also proposed [74,82]. 

Concerning the elution step, the nature of the elution solvent can be optimized to improve its efficiency and thus reduce the elution volume (which improves the enrichment factor) [84,86,89] or make it compatible with the analytical device used for toxin quantification. As an example, despite its efficiency, a glycine buffer was no longer used for the elution step due to its lack of compatibility with LC/MS–MS analysis [88]. 

As for conventional solid-phase extraction, the retention of an analyte on an IS during the percolation step depends on the volume of samples that is passed through the IS and the content of this sample [1]. Therefore, the nature of the solvent used to extract the toxins from the samples (cereals, food, etc.) may vary according to the sample matrix to ensure a good extraction yield [20], but it must also be compatible with the percolation conditions on the ISs since antibodies have a high affinity in aqueous media [89]. For polar toxins such as saxitoxins, the possibility to extract them from samples using phosphate buffer saline (PBS) solution constitutes a real advantage [101,114]. When solvents or hydro-organic mixtures are required, as often reported in Table 1 and Table 2, they can be either evaporated and the toxins next dissolved in water or PBS, or directly diluted with these aqueous solutions to decrease the solvent proportion that affects the retention on IS. However, with this second approach, the dilution rate affects the final sensitivity of the method and has to be carefully optimized [27,32,65,75]. Reported or calculated dilution factor values are mainly between 2 and 10. It is worthwhile noticing that a residual amount of solvent in the extract can sometimes be necessary to ensure the complete solubilization of the toxins [74,77,84,85,86,94,95,96]. Moreover, some authors suggested to optimize the extraction conditions of the toxins from the sample not only regarding the final extraction recovery of the toxins but also by studying the effect of the nature of the extraction solvent on the final selectivity measured by the removal of the interfering peak in the final chromatogram. As an example, for mycotoxin analysis, it was often mentioned that the addition of NaCl in the extraction solvent strongly contributes to the improvement in selectivity because it induces the precipitation of the proteins that are thus removed from the extracts [33,34,35,36,37]. The addition of a surfactant in the extract to be percolated was also reported to improve the clean-up effect as it contributes to limit nonspecific interactions of sample components with the IS [40]. Among the parameters affecting extraction recoveries, the pH of the sample was sometimes also studied [27,29,82,86,89]. 

As previously mentioned, an IS that contains several antibodies allows the simultaneous extraction of toxins from different chemical groups, thus decreasing both the global analytical time and the cost of the method as only one cartridge is required. Nevertheless, if the targeted toxins have different physico-chemical properties, it improves the difficulties to find the extraction conditions leading to high recoveries for all the toxins [42] without affecting the stability of some of them [48]. This may explain why some authors preferred to run the samples over several ISs [51] even if it means assembling the ISs in series for the elution step [61]. 

The volume of sample that can be percolated through an IS is limited by the affinity of the antibodies towards the antigen, as previously mentioned, but also by the number of antibodies immobilized, which defines the IS capacity that should not be overloaded. The capacity corresponds to the maximal amount of a target molecule that can be retained by the IS during the percolation step. It depends on the nature of the antibodies (mAbs or pAbs), of the grafting yield, and of the antibody accessibility for the antigen. This value can be provided by manufacturers, and values of about 1.4–1.6 µg of toxins were depicted for T2 or zearalenone affinity columns from Vicam for example [40]. 

To avoid the IS overloading, as can be seen from the data reported in Table 1, while the volume of sample/extract percolated is variable, the equivalent amount of sample contained in the percolation fraction never exceeds 1.5 g and is generally equal to or less than 0.5 g. This suggests that all commercial cartridges contain similar amounts of IS and therefore similar amounts of immobilized antibodies. Nevertheless, an easy way to determine the real capacity of an IS consists of measuring the amount of analyte retained as a function of the analyte amount present in the percolated sample. The amount of analyte retained by the IS can be determined by analyzing the elution fraction to measure the analyte amount that was fixed during the percolation and next desorbed applying the immunoextraction procedure. It can also be estimated by measuring the residual analyte amount in the percolating fraction after the percolation of a huge amount of toxin causing the overloading of the IS. Some data are presented in Table 2 for laboratory-made ISs. These values are difficult to compare because they are given in different units: per g or ml of sorbent, per number of antibodies, etc., but they are always in the range from the hundreds of ng to a few µg per gram or ml of sorbent [10]. These capacity values partly result from the grafting yields, some values being listed in Table 2 and being close to 100% for most of the reported studies. However, lower grafting yields may be obtained if steric hindrances occur during the grafting. This is why it can be interesting to optimize the number of antibodies for a given amount of sorbent as reported [69,70,79]. However, only a theoretical capacity can be calculated based on the grafting yield because the real capacity depends on the number of specific and active antibodies, which is unknown when using pAbs, and steric hindrances that could prevent the analyte from accessing the antibody recognition sites. The proportion of active antibodies can be deduced from the experimentally determined capacity value. As an example, values of 39% or 65% of active antibodies were reported for ISs prepared by the grafting of a poly(GMA-co-EGDMA) monolith [93] or sol–gel entrapment [73], respectively. Concerning laboratory-made ISs, there are only a few studies that give data about the repeatability of the preparation of ISs [71]. For an IS prepared by immobilizing antibodies on Sepharose, recoveries were found similar for the extraction of α- and β-amanitins on two independently prepared cartridges [87]. The column-to-column reproducibility was also determined by preparing nine sol–gel immunoaffinity columns on nine different days and mean recovery for DON was found to be 97.8% with a relative standard deviation (RSD) value of 1.4% thus indicating the high repeatability of this preparation method based on the entrapment of Abs in sol–gel [89]. For an IS prepared in a 100 µm i.d. capillary to be coupled on-line with nanoLC, the repeatability of the synthesis of monoliths estimated by the evaluation of their permeability was first studied and an RSD value of only 6.2% was obtained for three independent synthesis. After their grafting with antibodies, a mean extraction recovery of 73% was obtained for microcystin-LR with an RSD of 5.4% showing the similarity of the results obtained with these three ISs [71].

At last, commercially available ISs are not reused, thus explaining the use of pure methanol or sometimes acetonitrile as eluting solvent with the possible addition of up to 2% of acetic acid to increase the elution strength. Regarding the ISs prepared in laboratories, softer elution conditions are indicated in Table 2, such as the use of a water–acetonitrile or water–methanol mixture, to favor the reuse of the ISs. However, it is worthwhile to notice that the use of pure methanol does not prevent the reuse of ISs [86,88,94,95]. The reusability of ISs was not so much studied even for laboratory-made ISs, but some works demonstrated that ISs can be reused 5 [86,88], 6 [70], 8 [96] or even more than 60 times [90] without observing a decrease in the extraction recoveries. For a saxitoxin IS, the elution of this polar toxin was achieved by a glycine/HCl buffer that was selected because those mild conditions offered the possibility to reuse the IS up to 50 times the IS [101]. It was also reported that an IS prepared by Abs encapsulation in sol–gel can be reused 25 times and be stored at room temperature over 19 days in water or 20 weeks in PBS [89]. So, if leaching of antibodies can be a problem in sol–gel techniques because of the high porosity of the sol–gel matrix and the fact that Abs are not covalently bound, no, or negligible, leaching seems to be observed. Other studies carried out with commercially available ISs or laboratory-made ISs prepared by covalent bonding reported the possibility to store the ISs during either 360 days at 4°C or 30 days at room temperature [85]. Storage conditions are also given by manufacturers, such as, for example, the possibility to store ZearalaTest WB column 18 months at 4°C or 12 months at room temperature [40].

### 2.3. Immunoextraction Using Other Formats

In recent years, much research has been devoted to the development of miniaturized extraction devices with the aim of limiting reagent consumption and reducing sample volume [3]. Thus, for toxin analysis, as an alternative to conventional cartridges used in off-line mode and containing from 2 to 500 mg and sometimes up to 2 g of IS, as shown in Table 2, other formats have been proposed. Among them, microparticles and nanospheres were prepared and used for solid-phase extraction in dispersive mode (dSPE), also named immunocapture, mainly in the field of protein extraction. This dSPE mode was reported for 34% of the studies cited in Table 2. The particles were prepared by the covalent immobilization of antibodies on NHS-activated Sepharose beads [70], or tosyl-activated magnetic beads [102,110,111] or by non-covalent immobilization on protein G- [80,99,104,105,106,107,108] or streptavidin- [72,79,109] activated magnetic beads or on amino-coated hollow glass magnetic microspheres [100,101]. 

In dSPE, the extraction is carried out by introducing the sorbent directly in the sample instead of percolating the sample through a cartridge containing the sorbent. After a sufficient extraction time under stirring, the particles are recovered mainly by centrifugation or by a magnetic field (when using particles with a magnetic core) to be further introduced into a suitable desorption solvent. As for IS used in SPE cartridge, the nature of the extract put in contact with the IS particles, as well as the nature of the washing and elution solutions, rather called desorption solutions in dSPE, affects the extraction yields. In addition, it is necessary in this mode to optimize the extraction, the desorption times and the vortex speed. It appears that the extraction step takes from 1–10 min [70,72,79,80,100,101,104,105] to 1–2 h [99,102,107,108,110,111], the desorption step being carried out with a similar or shorter time. Most of the procedures include a washing step before desorption to ensure an optimal selectivity, but the duration of this step was never mentioned. The duration of the overall extraction procedure is therefore quite long, but only a small amount of phase is used, which reduces the costs of the device. Indeed, the polypropylene reservoir and frits, which can be clogged during the percolation of certain samples and thus requiring prior filtration, are no longer used. This certainly explains why most of the applications of ISs in dSPE mode concern protein toxins as illustrated in Table 2 that were often monitored in milk or plasma samples that contain huge amounts of other proteins that can clog frits. Indeed, dSPE was applied in 75% of the cases to these protein toxins in reduced sample volume by adding no more than 20 µL of beads in 500 µL of sample. Concerning the desorption of proteins, the addition of trypsin in the desorption solution was proposed to carry out simultaneously both desorption and digestion steps [102,104,105,111]. This allows us to reduce the overall duration of the analysis but hinders the reuse of the IS, as the antibodies are also digested by the protease, thus leading also to peptides that will make the analysis of the target proteins more complex. At last, similar to SPE cartridge that may contain several antibodies to trap, simultaneously, toxins from different classes, multiplex-immunoextraction of three different toxins was described by Dupré et al. who mixed three batches of beads, each batch being prepared with antibodies specific of one toxin [111].

In addition to the dSPE mode, IS particles can be packed in a small size precolumn (5–20 mm length and 1 to 4.6 mm internal diameter (i.d.)) connected to switching valves and an LC column. Different types of set-up exist for this coupling and they allow the automation of the whole analytical procedure [1]. This set-up at the conventional format was not described for toxin analysis but the integration of ISs on-line with the separation step was proposed under a miniaturized format thus requiring us to modify the way to prepare ISs. Indeed, in order to integrate the immunoextraction sorbents into miniaturized analytical methods, such as capillary electrophoresis (CE) and nanoLC, new approaches have been proposed that consist mainly in the in-situ synthesis of a monolith that are grafted in a second step with antibodies. This monolith must be hydrophilic to limit the contribution of nonspecific hydrophobic interactions during the extraction of the target analytes and must have an accessible function for antibody grafting. Such type of miniaturized ISs was recently reviewed [3] showing the growing interest for the miniaturization of ISs, but the development of monolithic ISs for toxin analysis is still reduced. One of the two reported works consisted of the in-situ synthesis of a 5 cm organic monolith on one end of a long silica capillary of 75 µm i.d. by radical polymerization using glycidyl methacrylate (GMA) as monomer and ethylene glycol dimethacrylate (EGDMA) as crosslinking agent [93]. The hydrophilicity of GMA, which possesses an epoxy group allowing antibody grafting, has often been advanced to justify its use in order to reduce the risk of non-specific interactions by limiting the hydrophobic effect. This device was applied to the extraction of OTA from pure spiked water samples before its elution by a solvent plug and its detection by laser-induced fluorescence detection (LIF) through the capillary. The second development concerns the preparation of a 5 cm hybrid monolith that was synthesized in a 100 µm i.d. capillary by hydrolysis and condensation of organosilanes and alkylethoxysilanes (by the sol–gel process) and used for the covalent grafting of anti-microcystin-LR antibodies [71]. The resulting monolithic IS was coupled on-line to nanoLC/UV via a nano-switching valve and applied to the analysis of microcystin LR in an algae extract. In this last case, a reduced sample volume of 150 nl was enough to determine microcystin-LR in the algae extract. For the poly-(GMA-EGDMA) monolithic-based IS, the amount of grafted mAbs was 18 mg/g thus allowing the retention of 1.2 pg/cm (3 fmol/cm) of OTA, which means that 39% of the randomly immobilized mAbs were active [93]. A higher capacity of 40 mg/cm (40.2 nmol/cm, 2.11 nmol/g) of MC-LR was reported for the hybrid monolithic-based IS [71]. This difference is mainly due to the higher specific surface area of hybrid than organic-based monoliths. This 40 mg/cm capacity of MC-LR corresponds to a binding density of 0.543 pmol/mL of active mAbs. This monolithic approach has also been used for the integration of ISs in chips but not yet applied for toxin analysis [3].

### 2.4. Potential of Immunosorbents for the Reliable Quantification in Real Samples

ISs constitute a good mean to concentrate the targeted toxin(s) while removing matrix effect thus allowing the analysis of the extract with simple and fast analytical methods adapted to numerous samples such as bioassays achieved in 96-well plates (ELISA or enzymatic inhibition assay) [16,73,74,77,98] as it was performed mainly for microcystin analysis. In addition, it was shown by Chiavaro et al. that the high selectivity of the IS allows the direct determination of AFs B_1_ and M_1_ at 1 µg/kg in pig liver extracts using only fluorescence detection [17]. Fluorescence was also directly applied to the analysis of OTA in wine, but it necessitates an additional step of purification of the IS eluate on amino silica [28]. However, in the majority of cases, as shown in Table 1 and Table 2, the ISs were applied upstream of liquid chromatography coupled initially mainly to fluorescence (LC/Fluo) for native fluorescent compounds or after post-column derivatization. The reliability of methods combining IS and LC/Fluo was proven by interlaboratory studies [12] or applications to certified reference materials, as it was carried out for T2 and HT-2 toxin analysis [34,37]. Comparison with the Association of Official Analytical Chemists (AOAC) method analysis was also performed showing the performance of the IS associated to LC/Fluo in terms of clean-up efficiency [39,41], but also by limiting solvent consumption, liquid–liquid extraction step being no more necessary as purification step as mentioned for ZON analysis [39].

Initially applied as a confirmatory method in the case of mycotoxins [19], the coupling of IS extraction with LC–MS is now unavoidable. This coupling has the advantage of being both more specific and applicable to a wide range of compounds. This constitutes a serious advantage when using multi-analyte ISs, but also adapted to new toxins such as protein toxins. It is well known that matrix effects can affect the sensitivity and accuracy of LC–MS/MS method. As such, it has been demonstrated by many authors that an IS clean-up can solve this problem by removing most of the interfering compounds from the final extract. Indeed, it was shown by Yue et al. that in contrast to conventional sorbents, ISs suppressed matrix effects for LC–MS/MS analysis of STX in bivalve extracts [114] thus allowing an external calibration. This simple calibration method was also applied to the quantification of T2 and HT-2 toxins in 20 different samples of food [38] or a mix of mycotoxins in different samples [55] as no matrix effects were observed using the IS. IS cleanup also helps the reliability of the LC–MS/MS analysis. As an example, Senyuva et al. reported that, in addition to the improvement in terms of sensitivity, peaks observed in LC–MS had Gaussian shapes and were essentially indistinguishable from standards [68]. There was also much closer agreement of ion ratios with standards when samples received cleanup. 

Nevertheless, despite the use of ISs, other authors mentioned that there are still some matrix effects that may affect the sensitivity and accuracy of LC–MS/MS. To circumvent the risk of false quantification in LC/MS caused by these matrix effects, a possibility is the use of matrix match calibration. It consists of using a blank extract of the studied matrix passed through the IS and spiked at different concentration levels to construct a calibration curve. This approach was also proposed by different groups to evaluate the clean-up effect of ISs [38,60,65,68,80]. Indeed, it was considered as necessary for the simultaneous quantification of HT-2 and T-2 toxins in maize and cherry samples [94], of several toxins in cereals [48,54,58,85], feed samples [84], or urine [62]. In return, matrix match calibration was studied and considered as not necessary for the analysis of a mix of mycotoxins in cereals [55,61] or SMC [31] in various samples thus allowing the use of the much simpler external calibration method. A similar conclusion was obtained for the extraction of OA from shellfish extracts in dSPE [80]. Indeed, in this study, a comparison was done between chromatograms obtained by injecting a mussel extract without any preparation (Figure 1A), with conventional SPE preparation (Figure 1B), and with IS preparation (Figure 1C) that shows the efficient removal of interfering compounds using the IS. The efficiency of the IS clean up was also demonstrated by the fact that the calibration curve for OA standard solutions prepared in methanol fits well with the curve obtained with OTA in scallop matrices (Figure 1D). This means that the matrix effects were minimal and that relatively accurate quantitative results could be achieved with external standard calibration curves [80]. 

For a study related to the simultaneous extraction of DON, ZON, T-2 and HT-2 toxins, matrix match calibration was only applied to the quantification of DON [67] as the quantification of the other toxins was not affected by matrix effects. The use of a “IS calibration standards”, as named by Vaclavikova et al., and prepared by spiking the elution solvent of the IS was also considered as efficient and less time consuming than matrix match calibration to correct the signal suppression or enhancement that occurs for some mycotoxins [60].

Thus, it appears that the conclusions differ between studies as to the potential of ISs to suppress matrix effects in LC–MS analysis. Indeed, this potential may also depend on the level of optimization of the extraction procedure, and in particular on the washing step which can efficiently remove residual interferents when perfectly optimized. It may also depend on the ionization capacities of the molecules in the source of the MS and on the level of sensitivity expected, the later point was illustrated by a study of Solfrizzo et al., showing that the choice of the calibration mode may depend on the contamination level. Indeed, for very low contamination levels they proposed to use labeled toxins (C13) to correct the quantification of some mycotoxins in cereals [50], while this was not necessary for higher levels of contamination. The use of isotopic dilution using C13-labeled mycotoxins is a good alternative to matrix match calibration that is quite tedious approach. It was also systematically used for the quantification of AF M_1_, and only to quantify AFs and OTA at low levels of concentration [52]. It was also applied to the quantification of SMC in various samples to correct matrix effects [115]. However, labeled toxins are very expensive and they are not available for all the studied toxins. By this fact, for the quantification of ZON, an analog of this compound was used as internal standard [53,66].

## 3. Molecularly Imprinted Polymers

An alternative to ISs consists in using molecularly imprinted polymers (MIPs) that are synthetic polymeric materials possessing specific cavities designed for a template molecule involving also a retention mechanism based on molecular recognition. They have the advantage of being synthesized in a few days. Their stability, ease of preparation, and low cost for most target analytes make them attractive for the extraction of different classes of compounds (pesticides, drugs, emerging contaminants, proteins, and natural products such as toxins) from various complex samples (environmental samples, food extracts, beverages, biological fluids, etc.) [6,7]. Numerous papers related to the development of MIPs for toxins are summarized in Table 3. Most of the reported studies describe the synthesis of an MIP and its application to the extraction of toxins from various types of samples, but some studies also describe the potential of some commercially available MIPs. More than half of the studies still include mycotoxins, for which MIPs are already on the market. Indeed, the company Merck (ex Supelco) and R-Biopharm propose MIPs for patulin (SupelMIP^®^, EASIMIP TM, respectively) and the company Affinisep (ex PolyIntel) provides MIPs for several mycotoxins such as for patulin, ZON, OTA, DON, but also, as for ISs, for the simultaneous extraction of fumonisins and ZON (AFFINIMIP^®^ Fumozon) or AF, fumonisins, OTA, DON, ZON, HT-2 and T-2 toxins (Multimyco AFFINIMIP^®^). Other MIPs were developed mainly for phycotoxins, such as some microcystins (MCs), domoic acid (DA), and some gonyautoxins (GTX).

The conditions of synthesis of MIPs prepared in the laboratory are described in Table 3. In most cases, their synthesis involves first complexing in solution an imprint molecule, i.e., the template, with functional monomers through non-covalent bonds, followed by polymerization of these monomers around the template using a cross-linking agent and an initiator. After polymerization, the template is extracted from the polymer by exhaustive washing steps in order to break the non-covalent interactions, thus freeing the binding sites, i.e., cavities, complementary to the template in terms of size, shape and position of the functional groups. Various polymerization methods can be applied to produce MIPs. They can be obtained by bulk polymerization, often chosen for its simplicity of execution. With this approach, a monolith is obtained which must be ground and sieved to obtain particles suitable for extraction devices (particle diameter from 25 to 100 µm or even 400 µm). After a sedimentation step to remove the finest particles, the particles of interest are then packed between two frits in disposable cartridges such as ISs. The non-regular shape of the particles obtained by this method constitutes a real limitation for the direct use of MIPs as selective stationary phase in LC or for their on-line coupling with LC as discussed latter. This explains why the polymerization of a layer of MIP on the surface of beads of well homogeneous and defined size was proposed. As shown by the synthesis conditions described in Table 3, this approach to get an MIP layer on the surface of magnetic or non-magnetic particles was the most reported method for developing MIPs for dSPE. Otherwise, other methods of polymerization were also used for MIP synthesis, such as precipitation polymerization that takes place in the presence of a larger amount of porogen than bulk polymerization and that allows generation of micro- or nanospheres. There is also the suspension/emulsion polymerization that consists of the introduction of the organic-based polymerization mixture as droplets into an excess of a continuous dispersion phase (water or perfluorocarbon fluids) by agitation. Each droplet acts as a mini bulk reactor to produce spherical beads in a broad size range from a few µm up to a few mm. At last, electropolymerization also allows production of a film of polymer on a surface with adapted monomers, i.e., pyrrole.

The choice of reagents involved in the synthesis of MIP must be judicious as it defines the subsequent properties of the binding sites. Generally, the molecule to be searched for in the samples is the one taken as template for MIP synthesis. It was the case for some studies reported in Table 3 such as the ones dealing with the development of MIPs for DON [129,130], MCs [136,137,138], OTA [140,141,147], T2 toxin [154], and ZON [158]. However, the use of a structural analog to create the cavities during the synthesis was reported in 75% of cases to produce MIPs for toxins. This approach, named dummy imprinting, avoids the risk of residual target toxins leaching from the polymer during its application to real samples that could cause erroneous results. It was particularly used for synthesizing MIPs for toxins, compared to other target molecules, as it also allows one to decrease the cost of the synthesis for very expensive standards of toxin and prevent the problem of exposure to toxic molecules during the synthesis. The analogs must be very close to the target toxin in terms of structure and chemical functionalities. For example, the use of 1-hydroxy-2-naphthoic acid as a model for mimicking citrinin makes it possible to use a molecule with a substructure (aromatic ring linked to a hydroxyl group and a carboxylic function in the ortho position) in common with citrinin that can play a role in its recognition [124]. This choice is not always easy to achieve as reported by Kubo et al. Indeed, they evaluated several commercial aromatic dicarboxylic compounds such as isomers of phthalic acid as templates to produce an MIP for domoic acid, before selecting the o-phthalic acid as template [125]. 

The monomer must be chosen according to the nature of the interactions it can develop with the template and according to its availability to limit the cost of the MIP synthesis. Many monomers are listed in Table 3. The most commonly used monomers are acrylic monomers, mainly methacrylic acid (MAA), which is capable of developing strong hydrogen bonds with oxygen-rich molecules such as aflatoxin, citrine, OTA, patulin and their analogs taken as a model. The basic vinylpyridine (VP) monomer has been selected for its properties to promote interactions with the domoic acid analog. These monomers were used in combination mainly with ethylene glycol dimethacrylate (EGDMA) as a cross-linking agent in a moderately polar and aprotic solvent, i.e., chloroform, toluene or acetonitrile, thus promoting polar interactions such as hydrogen bonds with toxins or their analogs. If only one condition of synthesis was reported in most of the studies cited in Table 3, some groups reported the screening of different conditions of synthesis; for example, Ao et al. screened both four dummy molecules and five different monomers for their ability to produce an MIP with a high retention capacity for domoic acid [128]. The use of two dummy molecules simultaneously to improve the extraction capability of an MIP for patulin [152] or of an MIP for gonadotoxins [133] was also reported. Several dummy molecules and two different porogens were also screened for obtaining an MIP selective to the β-N-methylamino-l-alanine neurotoxin [121]. In another study, two monomers and two porogenic solvents were evaluated for the synthesis of an MIP for citrinin [124]. The ratio between the different reagents giving rise to the most selective MIP for DA was also studied [127]. It is worthwhile to notice that if many studies involved the synthesis and next the characterization of different MIPs, a computational approach was also reported to select the best monomer before launching the synthesis procedure [120,129].

After their synthesis, MIP particles (30 mg to 450 mg) were packed in cartridges between frits for off-line SPE for most of the study reported in Table 3. If with ISs, aqueous samples or extracts constitute the major composition of the percolation fraction to ensure a high affinity of antibodies, i.e., a high retention of the target compounds during the percolation step, more of a variety of types of solvent can be percolated through the MIP depending on the nature of the interactions involved between the toxin and the monomer residues of the MIP. This is why pure or acidified organic solvents were sometimes used as diluted solvent of dry extracts before their percolation through the MIP [121,139,145,159]. 

To automate the system, MIP particles can also be packed in small size precolumns to be directly connected to an LC analytical column [124,142,155]. With this on-line set-up, compounds trapped on the MIP precolumn are directly transferred to the analytical column by the LC mobile phase. While performance can be similar to off-line mode in terms of clean up with the advantage of requiring a lower sample volume for the same sensitivity performance (the whole amount of extracted toxins is transferred to the analytical column in this case), the lack of homogeneity and the excessive particle size affect the quality of the coupling [155]. This problem can be overcome by introducing a loop between the MIP precolumn and the analytical column. This set-up allows one to desorb the toxin from the MIP by a volume of solvent that is transferred to the loop before being injected on the analytical column [140,141]. The limitation in this case is the compatibility of the desorption solvent with LC analytical conditions.

As for immunoaffinity extraction, MIP particles (10 to 200 mg) can also be dispersed in samples to carry out the extraction in dSPE mode. If some authors reported the use of small size particles down to 200–500 nm [127,130], the sample volume was in the same range as for off-line SPE: 1 to 50 mL for dSPE and 1 to 100 mL for SPE. Most of the MIPs used in dSPE were synthesized at a layer on the surface of magnetic particles to facilitate the extraction procedure by replacing the time-consuming centrifugation step by the use of a magnetic field to recover particles or the desorbed fraction. As for ISs used in dSPE, in addition to the sample volume, the sample pH and the washing conditions, it is also necessary to optimize the extraction and desorption times and the vortex speed. Extraction on MIP by partition of the toxins between the sample and MIP was also achieved by filling an envelope made of polypropylene membrane sheet with MIP particles (15 mg). This device, introduced during 30 min in the sample, allowed the extraction of OTA from coffee, grape juice and urine [143], a desorption time of 20 min (under sonication to facilitate the transfer of molecule) being reported. In order to miniaturize the extraction device, Luo et al. proposed to cover a SPME silica fiber by a thin film of MIP (MIP fiber of 125 µm diameter) to extract plant toxins in 30 min, the desorption being achieved in 8 min [153]. In the field of the miniaturization of MIP for toxin analysis, one can mention a work related to the on-line coupling of a capillary containing an MIP (100 µm i.d., MIP film produced at the surface of a monolith) with an LC capillary column (300 µm × 25 cm) [160]. The feasibility of this on-line coupling was demonstrated for the analysis of several aflatoxins but without any application to real samples.

The selectivity of an MIP towards an analyte results from the presence of specific cavities whose number and affinity towards the analyte can be evaluated by different methods. They generally consist of studying the interactions between the target analyte and the MIP in a solvent close to the one used during the synthesis in order to promote the same interactions as those established during the polymerization step. These studies are generally conducted in parallel on a non-printed polymer (NIP), used as a control sorbent as it is carried out using sorbent grafted with non-specific antibodies to evaluate non-specific interactions on ISs. An NIP is obtained with the same synthesis procedure as for MIP but in the absence of the template molecule. The NIP has the same chemical functions at its surface as the MIP but without having specific cavities. The strength of the interactions is therefore assumed to be higher on the MIP than on the NIP because the target analyte can be retained at different points (sum of interactions) due to its spatial and functional complementarity with the cavities. Most of the studies listed in Table 3 reported this comparison achieved by introducing a known amount of the toxin of interest in a vial with a given amount of MIP or NIP particles. Once the system has come to equilibrium, the amount of adsorbed toxin is deduced from the amount that remains in solution. The number of cavities in MIP is thus evaluated by the difference between the amounts adsorbed on both sorbents. 

Equilibrium batch rebinding experiments also allow determination of Kd values. As an example, values of 13.5 µmol/L [127] and of 9.46 µmol/L [126] were obtained for DA MIPs synthesized using 4-VP as monomer and EDGMA as cross-linker, but using different solvent as they were obtained by precipitation and bulk polymerization, respectively. However, it is sometimes difficult to deduce from this value the real affinity of the MIP for the target toxin in real media when it is measured in a solvent very different from the sample nature. This is why, to evaluate this affinity in conditions close to the extraction procedure, De Smet et al. measured a Kd value of 7 µmol/L for a T-2 toxin MIP in batch in a methanol/water mixture that corresponds to the composition of the solution used to dilute the cereal extracts before their percolation through the MIP [154]. 

In addition to a high affinity, a high selectivity is expected for the MIP toward the target toxin. In SPE mode, the selectivity of an MIP can be evaluated by percolating the toxin on the MIP and NIP in parallel and measuring the extraction yields on each sorbent. High recoveries are then expected on the MIP while low recoveries are expected on the NIP, indicating both the sufficient affinity of the MIP for the toxin and the high selectivity of the extraction procedure. For example, extraction recoveries of 107% and 2% were obtained on the MIP and NIP for OTA, respectively [145]. This particularly high difference indicates a very high selectivity of the MIP towards OTA. This MIP/NIP comparison constitutes also an easy mean to evaluate the MIP specificity that can be defined for antibodies by the capacity of the cavities to trap some structural analogs of the toxin [120,121,130,136,148,158,159]. For example, recoveries of 89% and 76% were obtained using the MIP for the extraction of BMAA and its structural analog 2,4-diaminobutyric acid (DAB), respectively, indicating a high affinity of the MIP prepared by dummy imprinting for both molecules. In addition, recoveries of 45% and 18% on the NIP for BMAA and DAB, respectively, indicated that the MIP procedure was particularly selective for DAB as it was only slightly retained on the NIP [121]. 

As a low recovery yield on NIP ensures a high selectivity, the MIP/NIP comparison can also be used to optimize the extraction procedure in real media, particularly the washing step, as shown by Urraca et al. for the extraction of citrine [123] or ZON [159] from cereal extracts. This optimization carried out in a real sample is particularly important because many studies showed that the extraction yields obtained for a procedure optimized in a pure medium can be affected by constituents of the sample leading to a decrease in extraction yields [121,131]. It was also reported that constituents of the sample can more strongly affect the retention on NIP than on MIP thus giving rise to a larger difference in recoveries between MIP and NIP and then to a higher selectivity [121]. Some groups also reported the optimization of the extraction procedure using real extracts and studying extraction recoveries and the removal of interfering compounds on MIP alone [117,144].

As ISs, MIPs possess a limited capacity that depends on the number of cavities created during their synthesis. As previously mentioned, batch experiments allow us to evaluate the number of cavities by comparing adsorbed amounts on MIP with the one adsorbed on NIP. However, as this evaluation is made in a pure solvent, this number of cavities cannot be directly linked to the real capacity of the MIP, i.e., the real number of cavities that contribute to the extraction of the target toxin in real samples whose composition may affect the interaction strength between the toxin and the MIP. To determine its real capacity, the best solution is to percolate increasing amounts of the toxin on the MIP (or to disperse MIP particles in samples containing increasing amounts of the toxin) until the sorbent overload is reached, while applying the optimized extraction procedure [145,149].

To evaluate their potential, MIPs were compared to conventional sorbents in terms of recovery [128,154]. As MIPs for toxins were developed later than ISs, the latter are then considered as selective sorbents of reference and some results obtained with MIP were also compared with those obtained with ISs. In most of the cases, this evaluation consisted of comparing the cleaning effect on the baseline of chromatograms obtained by LC analysis after extraction with the two types of sorbent [144,145,157]. The results were similar for all these studies with a slightly higher purification effect obtained with ISs, which eliminate more interferents and thus lead to a cleaner baseline than with MIP. Nevertheless, the latter remains effective since no interferent was co-eluted with the toxin of interest. An example of comparison is given in Figure 2 that corresponds to the LC-Fluo analysis of a corn extract purified on a commercially available IS (Figure 2A) or on MIP (Figure 2B) [157]. As mentioned by the authors, the impurity peaks in the chromatogram corresponding to the purification on the IS were significantly lower than those observed on the chromatogram obtained when using the MIP, but the peak height of ZON in the two chromatograms did not differ so much. 

If the clean-up effect of MIP appears to be lower than that of IS, it was nevertheless reported that MIPs allowed us to suppress matrix effects observed after the purification of an algae extract on a mixed mode sorbent for BMAA analysis [121] or for the extraction of gonyautoxins in sea water [135]. Nevertheless, as with ISs, matrix effects were still observed in some studies even after MIP purification [118,142], although low in one case [142] but it can be noticed that in these studies no washing step was implemented in the dSPE procedure. Moreover, in one case, the MIP extract was directly analyzed by fluorescence, i.e., without any possibility to distinguish the toxin from residual interfering compounds by their retention time [142]. However, several studies reported the necessity to use the matrix match calibration curve method or standard addition method for the quantification of the toxins despite the purification effect brought by the MIP [120,152,159], but, as for ISs, the choice of the calibration method certainly strongly depends on the level of concentration of the toxin that is to be reached, the effort on the optimization of the MIP procedure in real samples, and the sensitivity and specificity of the analytical method including its detection mode.

At last, as commercially available ISs, the commercially available MIPs are considered as single-use devices although several studies reported the possibility to reuse an MIP up to 80 times [141,159] and even 500 times for an MIP used on-line with LC [124], as well as the storage of MIPs being easier to manage than that of ISs because of the high chemical stability of these polymers in various solvents.

## 4. Oligosorbents

As an alternative to ISs and MIPs, it is possible to use oligosorbents (OSs), also called aptamer affinity columns (AACs). Aptamers are short, single-stranded oligonucleotides (DNA or RNA, usually 20 to 110 nucleotides in length) capable of binding to a specific molecule with an affinity that can be of the same order of magnitude as that of antibodies. An aptamer specific to a target is identified in vitro by an iterative selection process called SELEX (systematic evolution of ligands by exponential enrichment) performed on an initial mixture of a very high number of different oligonucleotides [161,162,163]. Most of the already identified specific oligonucleotide sequences are directed against large molecules such as peptides, proteins, nucleic acids and even bacteria, but also for a significant number of small molecules such as drugs, organic pollutants, and even inorganic ions [163,164,165]. In comparison with antibodies, aptamers offer several advantages as selective extraction tools. First, they can be prepared for toxic targets as well as for targets that do not induce an immune response in vivo. Their production at large scale with little batch to batch variation in activity was already demonstrated as their short regeneration time is within minutes [164], whereas antibodies need 1 or 2 days to recover their active conformation. Moreover, modifications can be introduced during their chemical synthesis to improve their stability or to facilitate their immobilization [166]. If their use in the field of biosensors has been widely described as still recently for mycotoxins [161,167] and marine (bio)toxins [8,9], the development of OSs for their use as selective extraction sorbents is quite recent but seems to be a very promising approach [168]. As illustrated by the works summarized in Table 4, the development of OSs for toxins mainly concerns two classes of mycotoxins, OTA and aflatoxins. This field was initiated by the pioneering work of Aguado et al. who first described a specific sequence for OTA that was quickly implemented for extraction purposes [169].

As can be seen from the data in Table 4, aptamers can be modified during their chemical synthesis to introduce a chemical group (as thiol or amine) or molecule (as biotine) at one end to facilitate their immobilization on different types of sorbents. Thus, as antibodies, biotinylated aptamers were immobilized by non-covalent binding on commercially available activated sorbents such as streptavidin-activated agarose gel or magnetic beads [170,171,175,178,184,189]. Amino-modified aptamers were also covalently immobilized on sorbents previously mentioned for the covalent grafting of antibodies such as CNBr- or NHS-activated Sepharose [174,175,178,179,180,181]. At last, the possibility to easily introduce amino or thiol groups at their end also facilitates their grafting on many other types of support such as monoliths [182,185,186,187,188] or nanoparticles [172,176]. 

Some of the works listed in Table 4 reported the use of a spacer to maintain the binding properties of the aptamer when bonded to a surface. Ali et al. reported that the use of a C12 spacer arm resulted in an OS with a higher capacity than when a C6 spacer arm was used [179]. In another study, the use of a C7 spacer arm gave better results than the C6 spacer arm in terms of extraction efficiency of aflatoxins. Both results were explained by higher aptamer grafting yields with the longer spacer arm [173]. 

Capacity values of 19 [178] and 24 nmol/g [179] for OTA OS and of 4.5 nmol/g [173] for AF B_2_ OS prepared using activated Sepharose gel were reported. These values in nmol/L that correspond to values between 6 and 10 µg/g are of the same order of magnitude as those previously mentioned for ISs. In return, values of 1–40 µmol/g were often reported for MIPs developed for different classes of molecule [190], thus indicating a higher capacity of MIPs compared to OSs and ISs. For AF B_2_ OS, a grafting yield of 17% was mentioned, 68% of the grafted aptamers being considered as active for the binding of AF B_2_ [173]. A grafting yield of 74% was also reported for AF aptamers grafted on silanized NPs [176]. For the study reporting the highest capacity of 24 nmol/g, the proportion of active OTA aptamers was estimated to be 37% of the total amount of aptamers introduced in the grafting solution [179]. Data of capacities are also available for OSs prepared using monoliths as solid phase. Their comparison necessitates some calculations as the data are given in ng of toxin, in µg/cm^3^ or in pmol/µL of OS capillary. Capacity values higher than 22 pmol/µL [185,187], of 50 pmol/µL [186], were obtained for OTA. These values are high enough considering the low sample volumes, i.e., the low amounts of toxin, treated with these OSs. For the OS with a capacity of 22 pmol/µL, a grafting rate of 98% was obtained [187]. These two values higher than those obtained for an IS prepared using a similar monolith [71] could be explained by the smaller size of the aptamers which could facilitate their grafting.

As for ISs and MIPs, OSs are generally packed between two frits into disposable cartridges (10–60 mg or 100–300 µL gel) as conventional sorbents for off-line SPE procedure. OSs can also be used in the dSPE mode, the extraction and desorption conditions being similar to those applied in off-line SPE. In this case, the extraction and desorption times have also to be optimized. Extraction times from 8 min to 1 h and desorption times between 10 and 30 min were reported. In order to reduce again the aptamer consumption while developing OS, several miniaturized devices that consist, as for miniaturized ISs, of the immobilization of aptamers on monoliths prepared in situ in capillary of 75–100 µm i.d. [182,185,186,187,188], allowing the injection of sample volume as low as 250 nL [187], as was described.

The binding of analytes to aptamers during the percolation/extraction step of the sample results from a good spatial complementarity between the toxin and its aptamer. So, it results from the sum of interactions between both entities whose nature and strength are defined during the selection of the aptamers in the selection buffer, also named binding buffer (BB). Therefore, to favor the retention of the toxin during the extraction step, the sample composition must be as close as possible to the one of the BB. This is why most of the extracts were diluted in the BB before their percolation through the OS. This dilution also allows us to limit the effect of organic solvents used for the treatment of solid samples although it was also reported that sample extracts containing up to 10–15% solvent could be applied to OSs without observing an effect on the trapping efficiency of the OTA OSs [177,183,191] or on AFs (B_1_,B_2_) OS [175].

As illustrated by the data listed in Table 4, the elution/desorption of toxins from the OSs was ensured by hydro-organic mixtures or pure solvents such as methanol. These conditions of elution were often combined for OTA OSs with scavenging agents such as EDTA, which is well-effective on OTA aptamers, as they are highly sensitive to the presence of cations that help to stabilize their intramolecular quadruplex structure. As aptamers are known to be particularly sensitive to temperature, the use of a high temperature of 90 °C was also proposed to elute ZON from an OS [189]. Nevertheless, these elution conditions must be selected according to the method used to immobilize the aptamers. As an example, 40% ACN can be passed through an OS cartridge prepared by covalent immobilization of the aptamers while only pure water can be applied when the same aptamers were bonded via non-covalent biotin/streptavidin interactions to be reused [178]. 

As for ISs or MIPs, non-specific interactions can occur between toxins and the sorbent used for the immobilization of aptamers but also with the nucleotide sequence. In order to investigate the contribution of nonspecific interactions in the retention mechanism on a given OS, different approaches have been suggested. It was proposed to compare the retention of the toxins on the OS with their retention on a non-grafted sorbent [173,181,185] and/or on an OS grafted with an oligonucleotide of almost the same length but which is non-specific to the toxin [173,182,187]. At last, it was proposed to compare the retention of the toxins on the OS with their retention on an OS prepared with a scrambled sequence that contains the same nucleotides as in the aptamer specific to the toxin but in random position (thus preventing the formation of the specific complex with the toxin) [175,178,183,186,187].

These control sorbents were also as helpful as NIP to optimize the washing procedure and then obtain the optimal selectivity [183,187]. As an example, Wu et al. showed that two washing steps were required for the removal of OTA from the control nanospheres while 90% OTA still remained on the nanospheres grafted with OTA aptamers [183]. As for the two other selective sorbents, the potential of OSs was demonstrated by comparing the baseline of chromatograms obtained by LC analysis after extraction on the OS or a conventional C18 silica sorbent, highlighting the potential of the OS to remove interfering compounds that co-elute with the target toxins using C18 silica [178,183]. As for MIPs, the comparison was also conducted with ISs that are considered as the selective sorbents of reference showing similar results for MIP evaluation with again a slightly higher purification effect obtained with ISs, which eliminate more interferents and thus lead to a cleaner baseline than with OS, but the latter remains again sufficiently effective since no interferent was co-elected with the toxin of interest [178]. The chromatograms presented in Figure 3 also illustrate the selectivity extraction provided by an OS [187]. They result from the injection of only 250 nL of a beer sample simply diluted by a factor of 2 with the BB and of BB spiked at the same concentration with OTA (300 ng/mL) on a monolithic OS OTA in capillary (70 × 0.1 mm i.d.) coupled on-line to nanoLC/Fluo (laser induced fluorescence). The baselines corresponding to the beer samples are as clean as for BB as other beer components were eliminated during the percolation and washing steps on the OS, whereas the OTA was, on the contrary, selectively retained on it. In addition, OTA was not retained on the control sorbent (grafted with non-specific aptamers), demonstrating the contribution of OTA aptamers to the selective retention process.

The most important difference between OSs and other sorbents, ISs and MIPs, is their affinity towards structural analogs that can be poor. Indeed, if the possibility to trap simultaneously AF B1 and AF B2 on an OS was demonstrated, AFs G1 and G2 were not retained by the aptamers in contrast with antibodies that trap the four analogs [175]. Moreover, it was reported that OTA aptamers have 100-fold less affinities for OTB than for OTA, while a difference of only three-fold is observed for anti-OTA mAbs [191]. Moreover, the hydroxyquinone metabolite of OTA, that can be trapped by an OTA IS, was not retained by OTA OS [179]. However, the possibility to trap several analogs of OTA by an MIP was also demonstrated [147]. Nevertheless, these results highlight the possibility offered by the SELEX process to determine a unique sequence of oligonucleotides that will adopt a conformation very specific for its target while removing oligonucleotides that could present an affinity for other structural analogs. In return, it is possible to adapt the SELEX procedure in order to select aptamers showing an affinity for different targets by using them alternatively during the selection. These possibilities offered by the SELEX technology cannot be achieved to the same extent with antibodies [164].

At last, the number of OSs developed for the selective extraction of toxins is still very limited as illustrated by Table 4, as OSs were only developed for OTA, AFs B_1_, B_2_, and M_1_ and ZON. To be efficient in terms of extraction recoveries, aptamers must have a high affinity towards the toxin, i.e., low Kd as antibodies. OSs were developed with aptamers having Kd of 0.36 µmol/L, 50–85 nmol/L and 41 nmol/L for OTA, AFs, and ZON, respectively. As low Kd values in the nmol/L range were already described for other toxins, such as ricin [192], MCs [193], FUM B1 [194], cholera toxin [195] or GTX 1 and 4 [196], and OA [197], one can expect the development of new OSs for toxins in the near future.

## 5. Conclusions

Due to their occurrence at the trace level in very complex samples, toxins required the development of very powerful tools for the sample treatment before their LC/UV, LC/Fluo or LC/MS analysis. While immunosorbents have proven to be effective in purifying samples prior to analysis, as evidenced by the number of commercially available ISs, the possibility offered today by MIPs and OSs must be taken into consideration. The selectivity brought by these three types of selective sorbents improves the reliability of LC analysis by removing matrix components, thus reducing the matrix effect. This improvement in selectivity compared to conventional approaches, which involve the use of several sample pre-treatment steps, is also a key point in the development of a robust and simpler analytical method. These similar performances in terms of selectivity make the choice between these three sorbents quite difficult. ISs remain the most widely used sorbents for toxin monitoring due to their commercialization for many toxins, which also attests to their performance in this field of analysis characterized by a very high level of sample complexity. Chemically produced and therefore, in many cases, faster and at lower cost, makes MIPs and OSs more attractive for new development than ISs if the antibody has to be developed. This chemical production also offers the possibility, by screening polymerization conditions or by playing on the selection process, to adapt the response of MIPs and OSs, respectively, to structural analogs that one wishes or does not wish to trap. However, it should be kept in mind that the development of an OS will be fast only if the aptamer sequence has been identified and, so far, the expectations at this level remain high. Concerning their use, the development of a selective extraction on MIP can be longer and the procedure often needs to be adapted to each type of sample and depends on the nature of the interactions developed and thus on the synthetic reagents used. In return, the extraction procedures on ISs and OSs consist essentially of privileging aqueous samples or extracts because of the strong affinity of antibodies and aptamers in these media. It is also for the same reasons that it is difficult to associate several MIPs developed for molecules with very different chemical functions for their simultaneous extraction, whereas it is quite simple for ISs and OSs. On the other hand, in case of high contamination, MIPs have a higher binding capacity.

Finally, whatever the sorbent selected, the efforts made within the framework of their miniaturization, which lead to a reduction in their implementation cost, testify to the growing interest in these new formats.

## Figures and Tables

**Figure 1 toxins-12-00795-f001:**
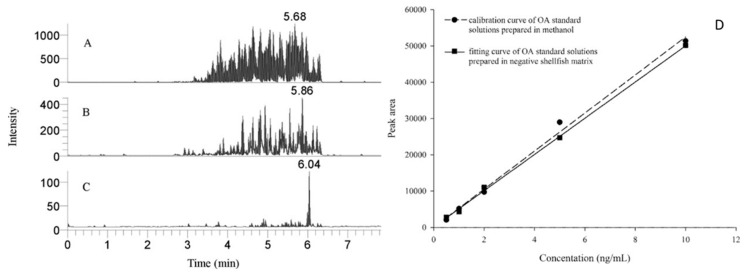
Total ion chromatograms of the selected reaction monitoring chromatograms of blank mussel samples without any preparation (**A**), concentrated through a conventional SPE preparation (**B**), and using an IS (**C**); comparison between the calibration curve for OA standard solutions and the fitting curve for standard-spiked samples (**D**). Reproduced from [80]. 2019, John Wiley and Sons.

**Figure 2 toxins-12-00795-f002:**
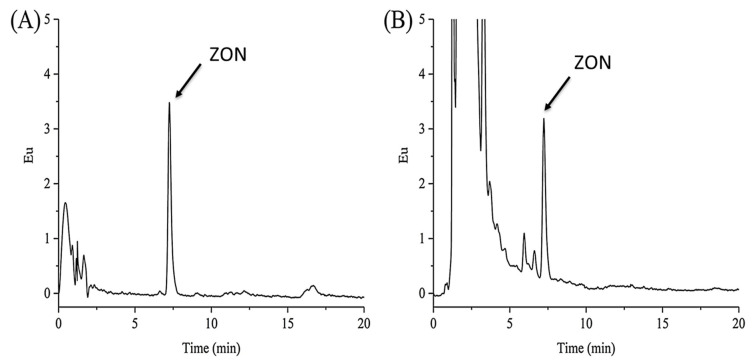
Chromatograms corresponding to the purification of a corn extract spiked with ZON on a commercially available IS (**A**) and on molecularly imprinted polymers (MIPs) (**B**). Reproduced from [157]. 2019, Elsevier.

**Figure 3 toxins-12-00795-f003:**
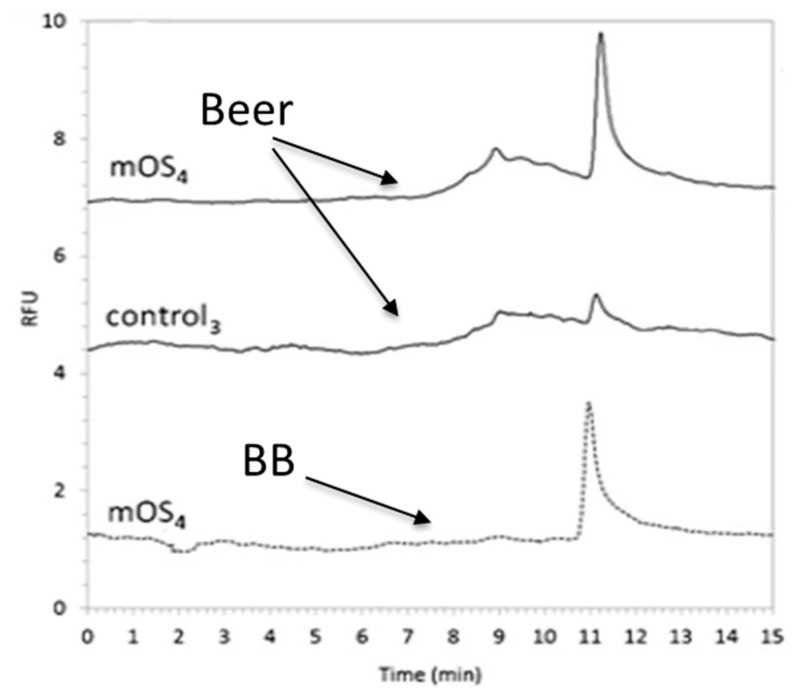
Chromatograms corresponding to the analysis of 250 nL of a beer sample (diluted in BB, 1:1) and BB spiked with 75 pg OTA extract on a miniaturized OTA OS and on the corresponding control OS coupled on-line with nanoLC-LIF. Reproduced from [187]. 2014, Springer Nature.

**Table 1 toxins-12-00795-t001:** Commercially available immunosorbents (ISs) for the analysis of single and multi-toxins in off-line solid phase extraction (SPE) mode.

Toxin(s)	Matrix	Marketed ISs (Company)	Extraction Solvent; Factor and Solvent of Dilution	V_sample_ (eq. of Solid Sample)	Washing	Elution	Analysis	Ref.
**Single analyte and analogs/metabolites**								
AFs (B_1_, B_2_, G_1_, G_2_)	Olive, peanut and sesame oils	Aflatest WB (Vicam)	MeOH/water 45/55; -, water	-	-	MeOH	LC/Fluo	[12]
	Nuts and based-nut products	Alfaprep (R-biopharm)	MeOH/water 7/3, NaCl; ×3, water	15 mL (eq. 1 g)	Water			[13]
	Baby food and feed	AlfaOchra HPLC^TM^ (Vicam)	ACN/water 78/22 (solid), ACN (milk); dried extract dil. in water	10 mL	Water	MeOH	LC/MS–MS	[14]
AF B_1_	Sidestream cigarette smoke	Aflatest P aflatoxin (Vicam)	-; ×4, water	20 mL	Water	ACN	LC/MS	[15]
	Organic spices and herbs	RIDA Aflatoxin column (R-Biopharm)	MeOH/water 7/3; ×4, water	1 mL (eq. 0.25 g)		MeOH	ELISA	[16]
AF B_1_ and AF M_1_	Pig liver	Afla^TM^ wide bore for M1 Aflatest-P for AFB1 (Vicam)	MeOH/water, NaCl; ×5, PBS, Tween-20 2%	20 mL (eq. 1 g)	PBS, Tween-20 2%	MeOH	Fluo, LC/Fluo	[17]
DON	Wheat	DON-Test HPLC (Vicam)	Water; -, -	1 mL (eq. 0.25 g)	-		LC/UV	[18]
DON, NIV	Rice, bran	DON NIV WB (Vicam)	Water, NaCl; ×5, PBS	10 mL (eq. 0.4 g)	PBS + water	MeOH + ACN	LC/UV, LC/MS	[19]
FUMs (B_1_, B_2_)	Cornflakes	Fumoni Test™ (Vicam)	ACN/MeOH/water 25/25/50; ×5, PBS	10 mL (eq. 0.4 g)	PBS	MeOH	LC/Fluo	[20]
OTA	Cereals	Easi-extract (Biocode)	MeOH/water 1/1; ×3, PBS	50 ml	Water			[21]
	Wine	Ochraprep (OP, Rhone Diagnostic Technologies) et Ochratest (OT, Vicam)	pH adjusted	10 mL + 10 mL PBS (OT) or 4 mL + 10 mL PBS (OP)	PBS + water (OP)			[22]
	Beer	Ochratest (Vicam)	Degassed; ×2, PEG - NaHCO_3_	10 mL	NaCl 2.5%, NaHCO_3_ 0.5% + water			[23]
	Urine		×2 (human) or ×3.4 (rat), NaHCO_3_ + filtration		Water			[24]
	Milk	Ochraprep (R-Biopharm) et Ochratest (Vicam)	-	50 mL				[25]
	Grapes, dried wine fruit, winery products	Ochratest (Vicam)	-, ACN/water or ACN/MeOH/water	-	Water, NaCl 2.5% + NaHCO_3_ 0.5%, PEG 1% + NaHCO_3_ 5%			[26]
	Ready-to-drink coffee		-; ×8, PB	5 mL	NaCl + NaHCO_3_ 0.5% + water + NH_4_CH_3_CO_2_	MeOH + AA 2%	LC/MS–MS	[27]
	Wine		-; ×2 PEG 8000 (1%), NaHCO_3_ (5%)	10 mL	NaCl, NaHCO_3_ + water	MeOH	Fluo, LC/Fluo,	[28]
	Cereals, spices		MeOH/water 7/3; ×1.8, water	40 mL	Water + PBS, Tween 20		LC/Fluo	[29]
OTA, OTB, α-OTA	Milk	Ochraprep (R-Biopharm)	LLE with CHCl_3_; back extraction with PBS	PBS extract	Drying			[30]
SMC	Cereal, cheese, beer	Easi-extract SMC (R-Biopharm)	ACN/water 8/2, NaCl; ×15, PBS	10–30 mL (eq. 0.25–0.5 g)	PBS + water	ACN	LC/MS–MS	[31]
T-2 toxin	Cereals	T2 TAG (Vicam)	MeOH/water 8/2; ×5, water	10 mL (eq. 1 g)	Water	MeOH	LC/Fluo	[32]
T-2 & HT-2 toxins		T-2 test (Vicam)	MeOH/water 9/1, NaCl; ×5, water	-				[33]
		Easi-extract T2 (R-Biopharm), T-2 Test HPLC (Vicam)	MeOH/water 9/1; -, NaCl 4%	-		MeOH (x 3, backflush)		[34]
		Easi-extract T2 (R-Biopharm)	MeOH/water 9/1 + 2% NaCl (oats)	-	-	-		[35]
	Chinese herbal medicines and related products	HT-2 HPLC (Vicam)	MeOH/water 9/1, NaCl; ×5, water	10 mL (eq. 0.5 g)	Water	MeOH	GC/ECD, GC/MS	[36]
	Oats	Easi-extract T-2 and HT-2 (R-Biopharm)	MeOH/water 9/1, NaCl; ×5, NaCl 4%	5–25 mL (eq. 1.5–0.3 g)	Tween 20 0.01% + water		LC/UV	[37]
	Food, Feed		MeOH/water 9/1, NaCl; ×5, water	25 mL (eq. 1 g)	Water		LC/MS–MS	[38]
ZON	Corn	ZearalaTest (Vicam)	ACN/water 9/1; ×10, water	10 mL			LC/Fluo	[39]
	Botanical root products, soybeans, grains, grain products extracted		MeOH/water 75/25; ×10, PBS, Tween 20 (0.5%)	50 mL (eq. 1 g)	MeOH/PBS, 15/85 + Tween 20 (0.5%) + water		LC/Fluo, LC/MS	[40]
ZON and metabolites (5)	Maize		ACN/water 9/1, NaCl; ×5, PBS, Tween 20 (0.1%)	10 mL	Water		Fluo or LC/Fluo	[41]
Multi-analytes								
AFs (B_1_, B_2_, G_1_, G_2_), OTA	Ginseng, ginger	AflaOchraTest (Vicam)	MeOH/water 7/3, 0.5%NaHCO_3_; ×5, PBS, Tween 20 (1%)	-	PBS + water + water/MeOH 85/15	MeOH	LC/Fluo, LC/MS	[42]
	Cereals	Aflatest and Ochratest (Vicam)	ACN/water 6/4 (OTA); MeOH/water 8/2 (AFs); ×5, PBS	50 mL (eq. 0.5 g)	Water	MeOH	LC/Fluo	[43]
	Sicilian sweet wines	Ochraprep, Easi-extract for AFs (R-Biopharm)	-; ×2 PEG 6000 (1%), NaHCO_3_ (5%)	20 mL	NaCl 2.5% + NaHCO_3_ 0.5% + water	MeOH/AA 2% (OTA), ACN (AFs)	LC/Fluo	[44]
	Meat products	Aflatest and Ochratest (Vicam)	MeOH/water 6/4, NaCl; × 2, water (AF) MeOH/water, NaHCO_3_ 1% 7/3; ×5, water (OTA);	10 mL (eq. 1 g)	Water (AFs); Tween 20/PBS (OTA)	MeOH (AFs)	LC/Fluo	[45]
	Spices and spices mixtures	AflaOchra HPLC (Vicam)	MeOH/water 8/2, NaCl; ×10, PBS, Tween 20	20 mL	Tween 20 0.01%, PBS + water	MeOH	LC/Fluo	[46]
	Ginger		MeOH/water 7/3, NaHCO_3_ 0.5%; ×4, PBS, Tween 20 (1%)	25 mL (eq. 0.3 g)	PBS + water		LC/Fluo	[47]
AFs (B_1_, B_2_, G_1_, G_2_), OTA, FUMs (B_1_, B_2_), DON, ZON, T-2 and HT-2	Maize	AOFZDT2^TM^ (Vicam)	Water (A) and then water /MeOH 3/7; PBS	Percolation of B and then of A	PBS (B), water (A)		LC/MS	[48]
AFs (B_1_, B_2_, G_1_, G_2_), OTA, ZON	Airborne from poultry house	AOZ (Vicam)	aqueous extract + NaCl; -, -	-	-		LC/Fluo	[49]
AFs (B_1_, B_2_, G_1_, G_2_), OTA, ZON, FUM (B_1_, B_2,_ B_3_), T-2 and HT-2	Cereals	AOF-MS-Prep and DZT-MS-Prep used in tandem	MeOH/water 7/3, NaCl; -, -	-	-	-	LC/MS–MS	[50]
AFs (B_1_, B_2_, G_1_, G_2_, M_1_), OTA	Turkish dairy food	3 ISs (no supplier mentioned)	MeOH/water 8/2, NaCl (AF B and G), 7/3 (OTA) and CHCl_3_, NaCl (AF M1); ×7, PBS (AF B/G, OTA), dried residue diluted in MeOH/PBS 2/98 (AF M1)	-	PBS (AF M1)	MeOH/water 1.25/1.75 (AFs B/G); MeOH/ACN 2/3 + water (AF M1); MeOH + water (OTA)	LC/Fluo	[51]
AFs (B_1_, B_2_, G_1_, G_2_, M_1_), OTA, DON, ZON, FUM (2), T-2 and HT-2	Food	AflaOchra Prep (R-Biopharm)	QuEChERSs method including LLE (hexane) to purify ACN extract; ×12.5, PBS	-	Water	MeOH	LC/MS–MS	[52]
AFs (B_1_, B_2_, G_1_, G_2_, M_1_), OTA, DON, ZON, NIV, FUS-X, VCG; T-2 and HT-2; CTN, 3-ADON,15-ADON, SMC	Food and feed extracts (84% ACN)	Mycosep 226 Aflazon + (COCMY2226, Romer labs)	-; ×2, ACN	-	-	-	LC/MS–MS	[53]
AFs (B_1_, B_2_, G_1_, G_2_), OTA, DON, ZON, FUM (3), T-2 and HT-2, NIV, 3-ADON, 5-ADON	Cereals	Myco6in 1 (Vicam)	Water + MeOH; ×3.5, PBS after partial evaporation	7 mL (eq. 0.5 g)	Water	MeOH + water	LC/MS–MS	[54]
AFs (B_1_, B_2_, G_1_, G_2_), DON, ZON, NIV, FUS-X, T-2 and HT-2, 3-ADON, 15-ADON, DAS	Corn, wheat, biscuit, cornflakes	Multisep 226 (Romer Labs)	ACN/water 85/15; none	10 mL	-	-	LC/MS	[55]
AFs (B_1_, B_2_, G_1_, G_2_), OTA, DON, ZON, FUMs (B_1_, B_2_), T-2 and HT-2, NIV	Spices, infant formula, coffee, nuts	AflaOchra Prep (R-Biopharm)	Water/ACN/AA 10/89.75/0.25 + salt (MgSO_4_/NaCl) + LLE (Hexane); ×25, PBS	50 mL	Water	MeOH	LC/MS–MS	[56]
	Corn and corn-derived products	Myco6in1 (Vicam)	MeOH/water 7/3; ×10, PBS	20 mL (eq. 0.5 g)	PBS + water			[57]
AFs (B_1_, B_2_, G_1_, G_2_), OTA, DON, ZON, FUMs (B_1_, B_2,_ ), T-2 and HT-2, NIV, 3-ADON	Cereal grains		ACN/water/AA 79.5/20/0.5; ×16, PBS	-		MeOH/water, 8/2, AA 0.5%		[58]
AFs (B_1_, B_2_, G_1_, G_2_), OTA, DON, ZON, FUM (B_1_, B_2_, B_3_), T-2	Herbs		MeOH/PBS 7/3 + LLE (hexane); ×26, PBS	-	NH_4_HCO_2_, FA (0.1%)	MeOH	LC/MS–MS, LC/HRMS	[59]
AFs (B_1_, B_2_, G_1_, G_2_), OTA, DON, ZON, FUMs (B_1_, B_2_, B_3_), T-2 and HT-2	Cereals, nuts		ACN/water/AA 79.5/20/0.5 + evaporation; -, PBS	10 mL (eq. 2.5 g)	Water		LC/MS–MS	[60]
AFs (B_1_, B_2_, G_1_, G_2_), OTA, DON, ZON, FUM, T-2 and HT-2	Cereals	AOF MS PREP, DZT MS-PREP (R-Biopharm)	MeOH/water 7/3, NaCl; ×13, PBS	20 mL (eq. 0.38 g)				[61]
AF M_1_, OTA, DON, DON analog, ZON (α,β), FUM B_1_	Urine	Myco6in1 (Vicam)	Oasis HLB SPE cartridge connected to the top of the IS; ×2, water	12 mL		MeOH + water		[62]
DON, ZON and 5 derivatives, 3-ADON, 15-ADON	Flour	Multi-IACs (Magnech Bio-Tech)	ACN/water 8/2; ×8, PBS	20 mL (eq. 0.25 g)	Tween 20 (1%) + Water	MeOH, AA 2%	LC/DAD	[63]
DON, ZON (+conjugated and metabolites)	Calf serum	DON Prep and DZT MS-Prep (R-Biopharm), NeoColumns for DON and for ZEN (Neogen), AokinImmunoClean C for DON and for ZEN (Aokin), Easi-extract ZEA (R-Biopharm)	Protein precipitation + drying; PBS, 5% MeOH	10 mL	Water	MeOH	LC/MS–MS	[64]
DON, ZON, HT-2 andT-2	Wheat, biscuit	DZT MS-PREP (R-Biopharm); MultiSep 226 (Romer Labs)	MeOH/water, 75/25; ×4, PBS, MeOH (15%)	5 mL (eq. 0.25 g)				[65]
DON, ZON, NIV, FUS-X, 3-ADON, T-2 and HT-2	Maize	Mycosep 226 and 227 (Coring systems Diagnostix)	ACN/water, 84/16; -	8 mL (eq. 2 g)	-	-	LC/MS	[66]
DON, ZON, T-2 and HT-2	Cereal and cereal-based samples		ACN/water, 85/15; -	5 mL (eq. 1 g)	-	-	LC/MS–MS	[67]
DON, ZON, T-2 and HT-2	Wheat, Maize	DZT MS-PREP (R-Biopharm)	ACN/water 8/2; ×40, PBS	8 mL	Water	MeOH		[68]

AA: acetic acid; CNT: citrinin; DAS: diacetoxyscirpenol; FA: formic acid; FUS-X: Fusarenon-X; LLE, liquid–liquid extraction; PEG: polyethylene glycol; PB: phosphate buffer (PBS: PB saline); PEG: polyethyleneglycol; VCG: verruculogen; ZER: zeranol. -: no data.

**Table 2 toxins-12-00795-t002:** Home-made ISs for the analysis of toxins.

Target Toxin(s)	Matrix	Extraction Solvent; Dilution Factor and Solvent	Sorbent, Amount of Abs	Grafting Yield or Density; Capacity	Extraction Mode	V_sample_ (eq. Sample Amount)/ Amount of Sorbent	Washing	Elution	Analysis	Ref.
**Toxins with MW < 1500**										
Bacterial toxin										
TTX	Marine organisms	MeOH, 1% AA; PBS (20% MeOH)	CNBr-Sepharose (0.5 g); mAbs (6 mg)	1106 ng/mL; -	Off-line SPE	25 mL (eq. 1 g)/0.5 g	Water	MeOH, AA (1%)	LC/MS–MS	[69]
Phycotoxins–Cyanotoxins										
Anatoxin-a	Pure water	-	NHS-Sepharose beads (27 µm, 10 µL); mAbs (100 µg)	-; 20 ng	dSPE	20 mL/ 10 µL	-	2-propanol	IMS	[70]
MC-LR	Algae extracts	-	Poly(APTES-co-TEOS) monolith; pAbs	-; 0.38 pmol (2.1 µg/g sorbent)	On-line SPE	150 nL/ 45 × 0.1 mm i.d. capillary	PBS	ACN/water (LC mobile phase)	Nano-LC/UV	[71]
	Urine	-	streptavidin-magnetic beads; Biotin-Abs		dSPE	100 µL	-	Water/ACN 7/3, FA (0.5%)	LC/MS–MS	[72]
	Pure water	-	Sol-gel entrapment (TEOS)	-; 4.28 µg	Off-line SPE	1 L (eq 2.5 g)/ 0.5 g	-	ACN/water 7/3	ELISA, LC/MS	[73]
MC-LR, MC-RR, MC-YR	Real waters	-	Glutaraldehyde-silica; purified pAbs	-; 1.8 µg/g IS		20 mL (0.5% MeOH)/ 0.25 g	Water + water/MeOH 8/2	MeOH/water 8/2	ELISA, PP2A, LC/MS	[74]
	Cyanobacteria, real waters	MeOH/water, 75/25; x0.75, PBS	pAbs	-		100 µL	PBS + water + MeOH/water 25/75	MeOH	LC/DAD, CE (MECK)	[75]
MC-LR, MC-RR, MC-YR, MC-LA	Algae and fish extracts, real waters	-; <15% MeOH	CNBr-Sepharose and silica; -	-		5–15 mL/ 0.1–0.2 g	PBS + water + MeOH/water 25/75	MeOH/water 8/2, AA (4%)	LC/UV	[76]
	Real waters and blue green algae extracts	-	Sepharose CL-4B; pAbs (1 mg/mg sorbent)	-; 0.2 µg		10 mL/ 2 mg	PBS + water + water/MeOH 85/15	MeOH/water 80/20, AA (2%)	ELISA, LC/UV	[77]
MC-RR, MC-YR, MC-LR, MC-AR, MC-FR, MC-WR, MC-LA, MC-LF, MCYST-LW and other MC variants	Real waters	Concentrated, filtered	CNBr-Sepharose and silica; pAbs	-; 200 ng/IS (Sepharose); 135 ng/IS (silica)		-	Water/MeOH 75/25	MeOH/water (+AA) 8/2	LC/DAD, LC/MS	[78]
	Urine	-	Streptavidin-beads (2.5 µL); biotinylated Abs (0.5 µg)	-	dSPE	500 µL/ 2.5 µL	-	Water/ACN 7/3, FA (0.5%)	PP2A (inhibition assay)	[79]
Phycotoxins—Diarrheic shellfish poisoning (DSP) toxins										
OA	Shellfish	MeOH, NaOH; dried extract in PBS	Protein G-magnetic beads; mAbs (1 mg/mg sorbent)	-	dSPE	1 mL/ 1 mg	PBS	MeOH	LC/MS–MS	[80]
	Algae extract	-; PBS/ACN 8/2	Silica; pAbs	-	Off-line SPE	-/125 mg	MeOH/water 3/7	MeOH-water 8/2	LC/Fluo, LC/MS	[81]
OA and derived form	Shellfish (hepatopancreas)	LLE; water/ACN 8/2	Glutaraldehyde-silica; pAbs	-; 16 µg/g IS		2 mL/ 125 mg	MeOH/water 7/3	PBS/ACN 7/3		[82]
OA, DTX-1 and DTX 2	Shellfish		-; Anti-AO mAbs	-		-	-	-	LC/Fluo	[83]
Mycotoxins										
AFs (B_1_, B_2_, G_1_, G_2_), OTA, ZON, SMC, T-2	Feed samples	ACN/water/AA 80/18/2; x3, PBS	CNBr-Crystarose; 4 mAbs (5 mg each/g sorbent)	-; 0.13 µg/mg Abs (sum of toxin)	Off-line SPE	10 mL (eq 0.6 g)/ 0.3 ml	PBS	MeOH	LC/MS–MS	[84]
AFs (B_1_, B_2_, G_1_, G_2_), OTA, ZON, T-2	Peanuts, corn, wheat	ACN/water/AA 80/19/1; x3, PBS (≤ 20% ACN)	CNBr-Sepharose (1.3 g); mAbs (20 mg each)	-; 9 µg/mL IS (sum of toxin)		10 mL/ 0.1 mL				[85]
AF B_1_	Cereals, peanuts, vegetable oils, Chinese traditional food	MeOH/water 6/4; x6, water (<10% MeOH)	CNBr-Sepharose (1 g, 3.5 mL); mAbs (9.92 g)	99.8%; 260 ng/mL		30 mL (eq. 1 g)/ 1 mL	Water		LC/MS	[86]
α- and β-Amanitins	Urine	Filtration; x1.8, PB	CNBr-Sepharose 2 mL; pAbs (6.4 mg)			9 mL/ 2 mL	PB + water + acetone/water 95/5	Acetone/ MeOH 1/1		[87]
DON	Cereals	MeOH/water 8/2; x2, PBS (<10% MeOH)	CNBr-Sepharose (1 g, 3.5 mL); mAbs (30 mg)	95%; 9.67 nmol/mL		10 mL (eq. 0.5 g)/ 1 mL	Water + MeOH/water 1/9	MeOH	LC/MS–MS	[88]
DON, 3-ADON, 15-ADON, deepoxy-DON	Foods, feeds (aqueous extracts)	-	Abs entrapped in silica gel (TMOS); mAbs	-; 1 µg/mg immob. Abs		/1 g	1% MeOH	ACN/water 4/6	LC/UV	[89]
FUMs (B_1_, B_2_) B_3_, OH-B_1_)	Dried feed samples	-; 100 µL for 10 mg, buffer	Protein A/PS-DVB (POROS); serum/mL sorbent	-		100 µL/ 30 × 2.1 mm column	-	Water/MeOH 7/3, FA (2%)	LC/MS	[90]
FUMs (B_1_, B_2_, B_3_)	Cereals	-	CNBr-Sepharose 4B (0.5 g); pAbs (1.27 mg/mL, 400 µL)	-		-	-	-	bioassay on cartridge	[91]
OTA	Beer	Degassed; x2, PBS, 1% PEG 6000	anti-IgG + CNBr-Sepharose (non-covalent bonding)	-		-	PBS, 0.05%Tween	OTA-HRP: competition		[92]
	pure media	None	polyGMA-co-EGDMA monolith in a capillary; -	260 ng Ab/cm; 1.2 pmol OTA/cm	In-line SPE	10 µL/ 8.5 cm × 75 µm i.d	PBS+ borate buffer	MeOH	CE/LIF	[93]
T-2, HT-2	Maize, cherry	MeOH/water 6/4; x6, water, ≤ 10% MeOH	CNBr-Sepharose (1 g); mAbs (30 mg)	-; 3 µg/mL IS (for each toxin)	Off-line SPE	30 mL (eq 0.5 g)/1 mL, 10 × 0.8 mm column	-	MeOH	LC/MS–MS	[94]
ZER + 3 analogs	Bovine muscle	MeOH; x5, PBS	CNBr-Sepharose 2 g; mAbs (50 mg)	96.3%; 2.7 µg/mL gel		25 mL (eq 2.5 g)/1 mL, 10 × 0.8 mm column	PBS + water + water/MeOH 7/3	MeOH	GC/MS	[95]
ZON, DON, T-2, HT-2	Grain products	-	CNBr-Sepharose (0.2 g); DON Abs (1.25 mg), H-2/HT-2 Abs (0.2 mg), ZON Abs (0.3 mg)	100%; 198–281 ng (for each compound)		-	Water or PBS	MeOH	LC/MS–MS	[96]
	Flour	-	Activated poly(GMA-co-DVB) µSpheres (0.3 g, 1 mL); DON Abs (1.25 mg), H-2/HT-2 Abs (0.2 mg), ZON Abs (0.3 mg)	-; 210–294 ng (for each compound)		/300 mg, 1 mL	-	-		[97]
ZON, T-2, HT-2	Feed samples	ACN/H_2_O, 8/2; x3, PBS	Anti-IgG-Sepharose (0.5 g, 1.8 mL); mAbs (ZON) and pAbs (T-2)	-		/0.2 g	-	-	bioassay, LC/MS–MS	[98]
Phycotoxins—Paralytic shellfish poisoning toxins										
STX	Human urine	-	Protein G-magnetic beads (30 mg/mL); mAbs, (1 mg/mL)	15 µg/mg (theory); -	dSPE	100 µL/1.5 mg	PBS + water	ACN/water 1/1, FA (2.5%)	LC/MS–MS	[99]
STX, NEO	Shellfish	-	NH_2_-coated hollow glass magnetic µSpheres; mAbs	5.8 mg/g; -		1 mL/25–100 mg	PBS	Glycine/HCl buffer	LC/Fluo	[100]
PSP toxins	Algae culture	PBS		5.5 mg/g; -						[101]
Protein toxins										
Abrin	Milk	-	Tosyl-activated magnetic beads (14.8 mg); mAbs against 4 epitopes (140 µg)	-	dSPE	500 µL/0.2 mg	PBS + Tween 0.05% + PBS + water	Trypsin digestion	LC/HRMS	[102]
*Androctonus australis Hector*	Venom		CNBr-Sepharose 2 g; purified pAbs (0.15 µmole)		Off-line SPE	/20 x1 cm column, 2 g	Tris HCl, NaCl	FA (pH 2.5), NaCl	UV	[103]
BoNT type A	Crude culture supernatant, food, environmental samples		Protein G-magnetic beads (3 µm), pAbs (BoNT A) and mAbs (ricin)		dSPE	500 µL/10–100 µL	HEPES	Trypsin digestion	LC/MS–MS	[104]
ETX	Milk, serum									[105]
Ricin	Pure media (buffer + BSA)					500µL/100 µL	Ammonium acetate (pH 4)	RNA incubation	LC/MS on adenine	[106]
	Milk					500µL/5 µL	Buffer	5% FA or 0.1% TFA in water	Tryptic digestion + MALDI-MS or/and LC/MS	[107]
	Milk, apple juice, human serum, saliva					500µL/20 µL	PBS, Tween + water	ACN, TFA		[108]
	Serum		Streptavidin-magnetic beads; biotinylated mAbs	55 µg/mg; 16.5 µg/mg		500 µL/20 µL	PBS + water	TFA 0.1%		[109]
Ricin, SEB, BoTN A and B	Milk, orange and apple juices		M-280 tosyl- paramagnetic beads (250 µL); mAbs	-		200 µL/8 µL				[110]
Ricin, SEB, ETX	Milk, human urine, plasma					1 mL/20 µL	PBS	Trypsin digestion	LC/HRMS (Q orbitrap)	[111]
Shigatoxin (protein) + analogs	Cell culture		CH-Sepharose 4B (2 g); purified pAbs (4 mg)		Off-line SPE	/2 g	Tris HCl + NaCl	Glycine (pH 2.7), NaCl 0.5 M	SDS Page	[112]
Staphilococcal enterotoxins A and E (proteins)	Dialyzed cell culture supernatant		Affigel 10 (agarose) 1 mL; mAbs (5 mg)			25 mL/1 mL	PB	AA, NaCl	UV	[113]

AA: acetic acid; APTES: aminopropyltriethoxysilane; BoTN: botulinium neurotoxin; CE: capillary electrophoresis; CNBr: cyanogen bromide; ETX: epsilon toxin; FA: formic acid, HRP: horseradish peroxidase; LLE, liquid–liquid extraction; NEO: neosaxitoxin; NHS: N-hydroxysuccinimide; PB: phosphate buffer (PBS: PB saline); PEG: polyethylene glycol; SEB: staphylococcal enterotoxin B; TEOS: tetraethoxysilane; TFA: trifluoroacetic acid; TTX: tetrodotoxin; ZER: zeranol. -: no data.

**Table 3 toxins-12-00795-t003:** Development and applications of MIPs for the selective extraction of toxins.

Target Analyte	Samples	Extraction Solvent; Dilution Factor, Solvent	MIP Synthesis: Monomer(s)/CL/Solvent; Polymerization Mode	Extraction Mode	V_sample_/ MIP Amount	Washing	Elution	Analytical method	Ref.
AFs (B_1_, B_2_, G_1_)	Maize	MeOH/water 7/3, NaCl; x3, PBS	MAA/EGDMA/EtOH SP on nanoporous carbon core	dSPE	-/ 80 mg	Water	ACN/water	LC/MS–MS	[116]
AFs (B_1_, B_2_, G_1_, G_2_)	Cereals	ACN/water 84/16; x12.5, PBS, Tween 20	MAA/EGDMA/MeOH; PP on mesoporous silica surface	Off-line SPE	50 mL/400 mg		MeOH	LC/Fluo	[117]
AFs (B_1_, B_2_, G_1_, G_2_, M_1_)	Fish, mussel liver	ACN/PB 6/4; x2, PBS	MAA/DVB/ACN; PP	dSPE	25 mL/40 mg	-	ACN/FA 2.5%	LC/MS–MS	[118]
Altenariol, altenariol monomethyl ether	Tomato juice, sesame oil	Water/ACN/salt; dry extract dil. in PB, 1% MeOH	4-VP, MA/EGDMA/ACN; SP on silica microspheres	Off-line SPE	25 mL/25 mg	ACN/water 5/95, ACN/ water 15/85	MeOH/TFA 99/1	LC/MS–MS	[119]
α-, β-amanitins	Human plasma	-	MAA,4-VP/EGDMA/DMSO; SP on vinylated silica microsphere	Off-line SPE/ dSPE	1 mL/1.3 g (SPE), 4 mL/20 mg (dSPE)	NaCl, PBS	MeOH	LC/UV	[120]
BMAA	Cyanobacteria	TCA; SPE (mixed-mode); ACN, 1% FA	APTES/TEOS/EtOH-water-HCl; BP	Off-line SPE	3 mL/25 mg	ACN/MeOH/ water 80/18/2		LC/MS–MS	[121]
Citrinine	Maize	MeOH/water 7/3	DMAEDM/TRIM/Acetone-ACN; BP		1 mL/300 mg	Water	MeOH/AA 98/2	LC/Fluo	[122]
	Rice	MeOH/water 7/3; dil. HEPES	4-VPU, MA/EGDMA/PVP-EtOH-water; SP on mNPs	dSPE	5 mL/ 200 mg	ACN/water 5/95	MeOH, TBA 50 mM	LC/UV	[123]
	Cereals, food supplement	MeOH; x2, water	AM/EGDMA/ACN; BP	SPE (on-line, 20 × 3 mm i.d.)	50 µL/25 mg	Water/MeOH 75/25, AA 0.5%	Water/ACN 7/3, 0.5% AA (LC mobile phase)	LC/Fluo	[124]
DA	Blue mussels	MeOH/water	4-VP/EGDMA/toluene; SP on polystyrene beads (5 µm)	LC (150 × 4.6 mm i.d.)	20 µL	-	ACN/water, 0.05%AA 7/3	UV	[125]
	Mussels		4-VP/EGDMA/toluene; BP	Off-line SPE	4 mL/150 mg	Water, ACN	MeOH, 5% AA	LC/UV, LC/HRMS	[126]
	Clams	ACN+ ACN/water 1/1; x2, water, 0.2% AA	4-VP/EGDMA/non-ionic surfactant; emulsion polymerization		1 mL/50 mg (200 nm)	-	ACN/citric acid 4/1	LC/UV	[127]
	Sea water and shellfish	MeOH/water 1/1; x1.66, HCl 0.1 M	TFMA/EGDMA/ACN; PP		500 mL (pH 1), 25 mL (shellfish extract)/100 mg	Water, FA 0.4%	MeOH, 1% FA	LC/UV	[128]
DON	Pasta	Water/EDTA	IA/EGDMA/DMF, BP		1 mL/100 mg	PBS	MeOH	LC/UV	[129]
DON, 3-ADON, 15-ADON, T-2, HT-2, FUS-X	Rice	MeOH/water, 7/3	MAA/DVB/ACN; SP on mNPs	dSPE	10 mL/30 mg (0,5 µm)	-	MeOH, 2% NaOH	LC/MS–MS	[130]
GTX 1,4	Sea water	-	MAA/EGDMA/DMSO; BP	Off-line SPE	1–50 mL/50 mg	AA 0.1 M	MeOH/water 95/5	LC/Fluo	[131,132]
	Microalgal culture	Water	MAA/EGDMA/CHCl_3_-PVA; suspension polymerization		1 mL/100 mg	MeOH/water 95/5, water	AA 0.1 M		[133]
GTX 2,3		AA; water			1 mL/200 mg	MeOH/water 98/2			[134]
	Sea water	-						LC/HRMS	[135]
MC-LR	Tap water, lake waters	x1.2, buffer	AMPSA, UAEE/EGDMA/DMSO; BP	Off-line SPE	3, 20 or 100 mL/10 or 30 mg	-	MeOH	ELISA	[136]
		-	Dopamine HCl/Tris; SP on magnetic GO			-	MeOH, AA	LC/UV	[137]
		-	MAA/EGDMA/toluene			MeOH/ water 1/9	MeOH, 5% AA	Bioassay	[138]
OTA	Red wine	C18 silica; MeOH extract	Acrylic monomers/EGDMA/CHCl_3_; BP	Off-line SPE	3 mL/100 mg	MeOH	MeOH, 2% AA	LC/UV	[139]
	Wine	Acidification (pH 1)	Pyrrole/EDMA/ACN; electropolymerization on stainless-steel frits	On-line SPE	100 µL	Water	MeOH, 1% TEA (pulse elution)	LC/Fluo	[140,141]
	Wheat	MeOH/NaHCO_3_ 3/7; x1.3, PBS, Tween 20	MAA/EGDMA/CHCl_3_; BP	On-line SPE (50 × 6.6 mm i.d.)	6 mL/45 mg	-	MeOH, 1%TBA		[142]
	Coffee, grape juice, urine	Water, 1% NaHCO_3_ (Coffee); x2 water (urine), pH 1.5	AFFINIMIP^®^ SPE Ochratoxin (Polyintel, AffiniSep)	dSPE (PP envelope)	10 mL/15 mg	Water	MeOH, 2%AA		[143]
	Beer, red wine, grape juice	-; x2, acidified water (pH 1)		Off-line SPE	20 mL	HCl 0.1 M/ACN 6/4		LC/Fluo; LC–MS/MS	[144]
	Wheat	ACN/water 6/4; x2, HCl 0.1 M			4 mL/50–100 mg			LC/Fluo	[145]
	Wine	Precipitation with PEG 8000	MAA/EGDMA/CHCl_3_; BP		2 mL/250 mg	Water /ACN 4/1	ACN, 2% AA		[146]
OTA, OTB, OTC	Rice, wine	ACN/water, 6/4 (Rice); dil. NaCl and NaHCO_3_ (wine)	Dopamine HCl; SP on mNPs	dSPE	50 mL, pH 3/15 mg		ACN		[147]
Patulin	Apple juice	-; x2, water, 0.2% AA	AM/EGDMA/ACN; SP on silica beads	Off-line SPE	2.5 mL/180 mg	Water, diethylether	Water, 1% AA	LC/UV	[148]
			MAL/EGDMA/ACN; SP on a silica-gel		1 mL/50 mg	NaHCO_3_, AA	ACN		[149]
		-; x2, water, 2% AA	Supel-MIP ^®^ SPE Patulin, EASIMIP TM Patulin, AFFINIMIP^®^ SPE Patulin	SPE (off-line, on-line)	4 mL (off line); 50 µL (on-line)/70–80 mg	Off-line: NaHCO_3_, water, drying, diethylether, drying; on-line: NaHCO_3_	Off-line: ethyl acetate; on-line: LC mobile phase (BF)		[150]
	Juices	LLE; dried extract dil. in ACN/acetate buffer (pH 4)	APTES/TEOS/MeOH-water; SP on activated silica beads (125–180 µm)	On-line SPE (15 × 4 mm i.d.)	50 mL/50 mg	-	LC mobile phase (BF)		[151]
	Juices, fruits	ACN, MgSO_4_, NaCl; dry extract dil. in water	MAA/TRIM/MeOH; PP	Off-line SPE	1 mL/30 mg	Water	MeOH	LC/MS–MS	[152]
Pyrrolizidine alkaloids	Herbal plants	DCM/MeOH, NaOH; dry extract dil. in water, 0.1% FA	Allylsulfonate/EGDMA/ACN; SP on silica fiber	SPME fiber	0.3 mL	MeOH, 0.1% FA	MeOH, NaOH	LC/MS	[153]
T-2 toxin	Maize, barley, oat	ACN/water 84/16 (+ LLE for oat); dry extract dil. in MeOH/water 2/8	MA/EGDMA/CHCl_3_; BP	Off-line SPE	1 mL/50 mg	MeOH/ water (6/4 -maize- or 2/8 – barley, oat-)	MeOH/AA 95/5		[154]
ZON	Beer	-	AFFINIMIP^®^ SPE Zearalenone (AffiniSep)	On-line SPE	50 µL	ACN/water 1/9 + AA 2%	LC mobile phase (ACN/water 35/65)	LC/Fluo	[155]
	Cereals	ACN/water 8/2, dil. water 0.2% FA	4-VP/EGDMA/dibutyl phtalate; SP on mNPs	dSPE	10 mL, ACN 2%/100 mg	Water	MeOH x3, ACN x2	LC/MS–MS	[156]
		Water extract	MAA/EGDMA/EtOH; PP with MOF	Off-line SPE	10 mL/100 mg		ACN/ water 9/1	LC/Fluo	[157]
	Seed-strain	Extract with 60% ACN	1-ALPP/TRIM/ACN; PP		1 mL/100 mg	ACN/water 7/3	MeOH/AA 95/5	LC/MS–MS	[158]
ZON, alpha ZOL	Cereals, swine feed	Dried extracts dil. ACN	1-ALPP/TRIM/ACN; BP		5 mL/450 mg	Water	MeOH/phosphoric ac. 95/5	LC/Fluo	[159]

AA: acetic acid; AM: acrylamide; ALPP: allylpiperazine; AMPSA: 2-acrylamido-2-methyl-1-propanesulfonic acid; APTES: aminopropyltriethoxysilane; BMAA: β-N-methylamino-L-alanine; BF: backflush; BP: bulk polymerization; DMAEDM: dimethylaminoethyl methacrylate; DMF: dimethylformamide; DMSO: dimethylsulfoxyde; DVB: divinylbenzene; EGDMA: ethylene glycol dimethacrylate; FA: formic acid;GO: graphene oxide; IA: itaconic acid; LLE: liquid–liquid extraction; MA: methacrylamide; MAA: methacrylic acid; MAL: maleic acid; mNP: magnetic nanoparticles; MOF: metal organic framework; PB: phosphate buffer (PBS: PB saline); PP: precipitation polymerization; PVA: polyvinyl alcohol; PVP: polyvinylpyrrolidone; SP: surface polymerization; SPME: solid-phase microextraction: TBA: tributylamine; TEA: triethylamine; TEOS: tetraethoxysilane; TFA: trifluoroacetic acid; TFMA: trifluoromethylacrylic acid; TRIM: trimethyltrimethacrylate; UAEE: urocanic acid ethyl ester; VP: vinyl pyridine; VPU: vinylphenyl urea.

**Table 4 toxins-12-00795-t004:** Development and applications of oligosorbents for the selective extraction of toxins. -: no data.

	Aptamer Modification/Spacer	Sorbent	Matrix	Extraction Solvent; Dilution Factor and Solvent	Extraction Mode	V_sample_/Amount of OS	Washing	Elution	Analytical Method	Ref.
AF B_1_	biotin/-	Streptavidin-coated microtiter plate	Corn	Dried extract in BB	Partition	500 µL	-	MeOH/water 8/2	LC/Fluo	[170]
		Magnetic streptavidin-NPs	Peanut oil		dSPE	-	-	-	Fluo	[171]
AF B_2_	-	Magnetic NH_2_-NPs								[172]
	NH_2_/C7	CNBr-Sepharose	Peanut	MeOH/water 7/3, NaCl; ×10, BB	Off-line SPE	5 mL/60 mg	BB	MeOH	LC/Fluo	[173]
AFs B_1_, B_2_		NHS-activated Sepharose	Peanut, maize, wheat, rice	MeOH/water 7/3, NaCl; ×9, BB		30 mL/300 µl				[174]
	Biotin-Apt	Magnetic streptavidin-agarose beads (28 µm)	Maize	MeOH/water 7/3, NaCl; dry extract in BB	dSPE	1 mL/200 µL		BB/MeOH 2/8		[175]
AF M_1_	-	Magnetic silanized NPs	Milk	LLE (hexane, MeOH, water); -		15 mL/8 mg	MeOH	DCM, AA		[176]
OTA	-	DADPA	Wheat extract	MeOH/water 6/4; x4, BB	Off-line SPE	10–12 mL/300 µL	BB	Tris-HCl, EDTA, 20% MeOH	Fluo	[169]
								MeOH/BB (2/8)	LC/Fluo	[177]
	NH_2_/C6	CNBr-Sepharose	Red wine	pH adjusted		1 mL/35 mg	BB/ACN 9/1	Water/ACN (6/4)	LC/Fluo	[178]
	NH_2_/C12		Wheat extract	ACN/water 6/4; ×10, BB				Water/ACN (6/4)	LC/Fluo	[179]
	NH_2_/C6	NHS-activated Sepharose	Ginger powder, TCM	ACN/water 6/4; ×10, BB		2, 3 mL/200 µL	BB	MeOH	LC/Fluo LC/MS–MS	[180] [181]
	SH/C6	Au NPs/POSS-PEI monolith	Pure sample	-		20 µL/100 × 0.1 mm i.d. capillary		Tris-HCl, EDTA	LC/Fluo	[182]
	NH_2_/C6	Magnetic carboxylated-NPs (100–250 nm)	Food	MeOH/water 5/5; ×2, Tris-HCl, Mg^2+^	dSPE	100 µL/100 µL	Tris-HCl	ACN/water (95/5) + 1% AA	LC/Fluo	[183]
	Biotin-Apt	Magnetic streptavidin-MOF	Corn, peanut	dry extract in HEPES, Mg^2+^		-/10 mg	-	-	LC/MS–MS	[184]
	SH/-	POSS-acrylate-based monolith (one-pot synthesis)	Beer	pH adjusted; ×2, BB	On-line SPE (loop)	100 µL/100 × 0.1 mm i.d. capillary	BB	ACN/Tris-HCl, EDTA (3/7)	LC/Fluo	[185]
	SH/C6	Poly(TMOS-co-MPTMS) monolith modified with AuNPs (25 nm)	Beer, wine	pH adjusted; ×9, BB		20 µL/100 × 0.075 mm i.d. capillary		ACN/Tris-HCl, EDTA (7/3)	LC/Fluo	[186]
	NH_2_/ C12	Poly(APTES-co-TEOS) monolith	Beer	×2, BB	On-line SPE	250 nL/ 70 × 0.1 mm i.d. capillary		nano-LC mobile phase	NanoLC/LIF	[187]
	SH/-	Poly(TMOS-co-MTMS) monolith	Beer and white wine		In-line SPE	768 nL (beer), 5 µL (wine)/ 15 × 0.075 mm i. d. capillary		ACN/Tris, EDTA, 3/7	CE/LIF	[188]
ZON	Biotin-Apt	Magnetic streptavidin beads	Beer		dSPE	-	-	90 °C, BB	Fluo	[189]

AA: acetic acid; APTES: aminopropyltriethoxysilane; BB: binding buffer; DADPA: diaminodipropylamine; (m)NP: (magnetic) nanoparticles; LLE: liquid–liquid extraction; NHS: N-hydroxysuccinimide; CNBr: cyanogen bromide; MPTMS: mercaptopropyl trimethoxysilane; MTMS: methyltrimethoxylsilane, SH, Thiol; TCM: traditional Chinese medicines; TEOS: tetraethoxysilane; TMOS: tetramethoxysilane; POSS*:* polyoctahedral silsesquioxanes.

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
