# Peer review of "Immunoaffinity Extraction and Alternative Approaches for the Analysis of Toxins in Environmental, Food or Biological Matrices"

_toxins, 2020, doi:10.3390/toxins12120795_

Round 1

Reviewer 1 Report

Some phrases are too long and need some clarification (i.e. the phrase from rows 269 - 273). Also there are some unclear phrases, i.e. from rows 175 or 247-249).

I recommend to break the long phrases in shorter sentences.

Author Response

We thank this referee for his careful reading of this review.  The sentences mentioned have been corrected and other sentences were modified to improve their understanding (Lines 172­-174, 179, 253-256, 279, 397-403, 477, 524-525, 615).

The manuscript has also been entirely re-read in order to reduce the length of some sentences (lines 35, 48, 81, 85,97, 109, 181, 266, 281, 288, 296, 312, 339, 345, 354, 370, 384, 467, 614, 645, 736, 739)

L’email a bien été copié  

Reviewer 2 Report

The review work carried out by the authors is clearly stated and well structured. An extensive bibliographic review has been carried out, including almost 200 citations. However, the conclusions could be improved by including a critical view from the authors point of view. I also suggest to the authors, if they consider it feasible, to include a final comparative table with the most important advantages and disadvantages of each of the purification / preconcentration systems.

Author Response

We thank this referee for his comments. Although a table allowing the comparison of supports was recommended, we prefer to limit the already high number of tables. Nevertheless, elements of comparison have been added in the conclusion to expand it as requested.

" These similar performances in terms of selectivity make the choice between these three sorbents quite difficult. IIs remain the most widely used sorbents for toxin monitoring due to their commercialization for many toxins, which also attests to their performance in this field of analysis characterized by a very high level of sample complexity. Chemically produced and therefore, in many cases, faster and at lower cost, makes MIPs and OSs more attractive for new development than ISs if the antibody has to be developed. This chemical production also offers the possibility, by screening polymerization conditions or by playing on the selection process, to adapt the response of MIPs and OSs respectively to structural analogs that one wishes or does not wish to trap. However, it should be kept in mind that the development of an OS will be fast only if the aptamer sequence has been identified and so far the expectations at this level remain high. Concerning their use, the development of a selective extraction on MIP can be longer and the procedure often needs to be adapted to each type of sample and depends on the nature of the interactions developed and thus on the synthetic reagents used. In return, the extraction procedures on ISs and OSs consist essentially of privileging aqueous samples or extracts because of the strong affinity of antibodies and aptamers in these media. It is also for the same reasons that it is difficult to associate several MIPs developed for molecules with very different chemical functions for their simultaneous extraction whereas it is quite simple for ISs and OSs. On the other hand, in case of high contamination, MIPs have a higher binding capacity.

Finally, whatever the sorbent selected, the efforts made within the framework of their miniaturization, which lead to a reduction in their implementation cost, testify to the growing interest in these new formats. "

L’email a bien été copié

Reviewer 3 Report

Immunoaffinity extraction and alternative approaches for the analysis of toxins in environmental, food or biological matrices is a relevant review on Immunoaffinity extraction approches. Authors reported 197 articles published generally during last 10 years.

Immunoaffinity sorbents, Molecularly imprinted polymers, Oligosorbents were the principal approach developed in this paper.

The content of this work could be helpful for scientific community, principally PhD students

comments

37 38: LC-MS suffers from matrix effects during the ionization step that 37 can lead to erroneous quantification: I propose “impact method sensitivity”

86 87: but their production suffers from a lack of reproducibility 86 in terms of time of response of an animal, of quantity and even of specificity : I propose to add « ethic issues »

Paragraph 2.1. Antibody production and development of immunosorbent : please add relevant references

757: 4. Conclusions: it must be n° 5 (5. Coclusion)

Author Response

We thank this referee for his comments.

Concerning matrix effect, as they can lead to signal enhancement or suppression, they have an impact on the sensitivity but especially on the reliability of quantification, whatever the concentration level. This is why we propose to keep the sentence referring to quantification.

“ethic issues“ was added in the sentence.

Two references were added in the paragraph 2.1.

The numbering of the conclusion was changed.

L’email a bien été copié